# MIMIC-VQA: Compiling Agentic Reasoners into Efficient Document VQA Models

## Abstract

Document Visual Question Answering systems face a fundamental architectural dichotomy: modular agentic frameworks decompose problems into interpretable subtasks but incur prohibitive inference latency through sequential tool orchestration, while monolithic end-to-end models achieve computational efficiency at the cost of reasoning transparency and spatial grounding capabilities. We present MIMIC-VQA, a knowledge distillation framework that transcends this trade-off by compiling the procedural reasoning of expert agents into efficient neural architectures. Our approach operates through a two-phase paradigm: first, a teacher pipeline orchestrated by Llama 4 Scout generates 102,447 Chain-of-Thought reasoning traces that explicitly encode multi-step problem decomposition, contextual retrieval, and deterministic spatial grounding; second, these traces train a pruned 9B-parameter student model derived from Gemma 3-27B to replicate the complete reasoning process—including intermediate steps and bounding box coordinates—within a single autoregressive generation. This procedural distillation enables the student to internalize the teacher's tool-based reasoning methodology while eliminating runtime dependencies on external components. Empirically, MIMIC-VQA achieves SoTA performance across DocVQA (89.7 ANLS), VisualMRC, FUNSD, and CORD benchmarks, demonstrating 20-30 point improvements in spatial grounding (mAP@IoU) over existing methods while operating 5.3× faster than the teacher system. The framework maintains 98.3% of teacher accuracy despite 66% parameter reduction, validating that complex multi-agent reasoning can be successfully compiled into compact neural representations. By treating sophisticated agentic systems as data generators rather than deployment models, MIMIC-VQA establishes a practical paradigm for scaling document understanding capabilities without prohibitive infrastructure costs. The dataset of reasoning traces and the official implementation are publicly available at: https://anonymous.4open.science/r/MIMIC-B5DF.

## 1 Introduction

Document Visual Question Answering (VQA) demands a robust integration of text comprehension, layout analysis, and spatial reasoning to interpret complex documents. While recent advancements like LayoutLMv3 (Huang et al., 2022), LayoutLLM (Luo et al., 2024), and DocLayLLM (Liao et al., 2025) have pushed the boundaries of textual accuracy, they often fail to provide precise spatial grounding for their answers. This limitation is compounded by standard evaluation metrics such as Average Normalized Levenshtein Similarity (ANLS) (Yujian & Bo, 2007), which measure textual correctness but overlook spatial accuracy. As a result, these models can generate plausible but unlocalizable answers, creating a "trust gap" that hinders their adoption in high-stakes applications where verifiability is critical.

This gap has created a dichotomy in system design. On one side, modular agentic frameworks (Shen et al., 2023; Castrejon et al., 2024; Han et al., 2025) produce highly accurate and auditable results by decomposing problems into a sequence of tool-based steps. However, their sequential nature incurs significant inference latency, rendering them impractical for large-scale or real-time use. On the other side, fast monolithic models (Mohammadshirazi et al., 2024) perform inference in a single forward pass but often operate as opaque "black boxes," lacking the interpretable reasoning of their agentic counterparts. This forces a trade-off between the speed of monolithic models and the trustworthy, step-by-step reasoning of modular agents.

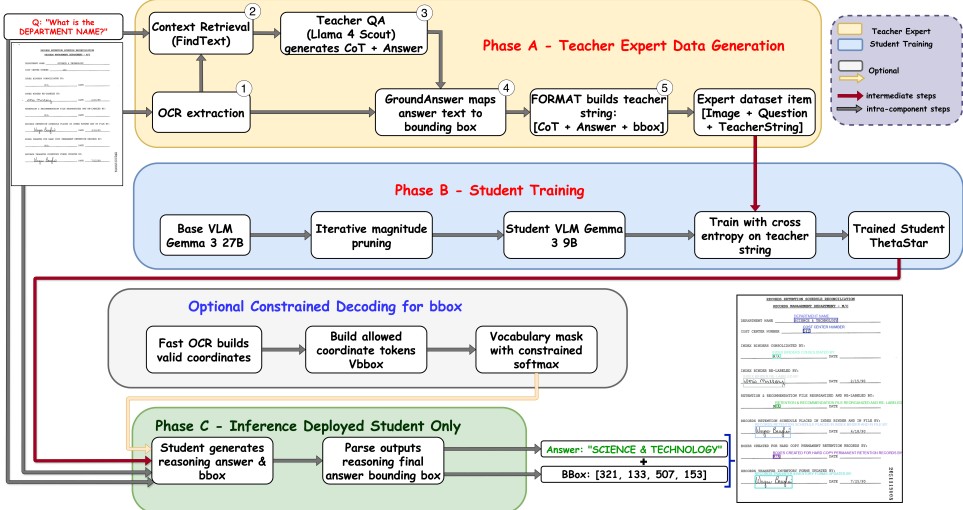

Figure 1: The MIMIC-VQA Framework. **Phase A (Teacher Expert Data Generation, yellow)** decomposes document VQA into modular steps: OCR extraction, context retrieval, teacher QA, deterministic grounding, and formatting a full reasoning string with answer and bounding box supervision. **Phase B (Student Training, blue)** prunes a large base VLM (Gemma-3-27B) to an efficient 9B-parameter student and distills it on the expert traces via cross-entropy. **At inference (Phase C, green)**, the student alone generates a chain-of-thought, final answer, and spatial coordinates in one pass. **An optional constrained decoding module (gray)** uses a lightweight OCR pass to restrict the vocabulary during <bbox> generation, ensuring robust coordinate outputs. The end-to-end process yields a compact student model capable of reliable reasoning and grounding on document VQA tasks.

Motivated by these challenges, we propose a novel framework that bridges this divide. Our central hypothesis is that the slow, explicit, and interpretable reasoning of an expert modular agent is itself a form of procedural knowledge that can be learned. We introduce **MIMIC-VQA** (Modular Imitation for Multimodally-Integrated Comprehension in Visual Question Answering), a teacher-student paradigm that effectively "compiles" the multi-step reasoning of a slow but accurate agent into a single, fast, end-to-end model. Unlike conventional knowledge distillation, which typically focuses on output logits, our approach distills the entire reasoning process.

Our approach begins with a "teacher" pipeline, where a Llama 4-based planner Meta AI (2025) orchestrates multiple tools to generate a gold-standard dataset of over 100,000 document VQA reasoning traces. Each trace contains a detailed Chain-of-Thought (CoT) that leads to the final answer and its precise location. We then train a compact "student" model—a pruned Gemma 3-27B Team et al. (2025) —to imitate this entire reasoning trace. The student learns to generate the full CoT, including the answer's bounding box represented as a sequence of text tokens (e.g., [450 80 120 25]), in a single, efficient forward pass. At inference time, the complex teacher agent and its tools are discarded, leaving only the highly efficient student model.

Building on this foundation, our contributions are as follows:

1. A novel teacher-student framework that successfully distills the complex, multi-step reasoning of a modular document agent into a single, efficient end-to-end Visual Language Model (VLM).

2. A methodology for teaching a VLM to perform precise spatial localization by representing bounding boxes as a textual sequence embedded within a generated CoT reasoning trace.

3. The creation of a high-quality dataset of over 100,000 document VQA reasoning traces, which we plan to release to the community to foster further research.

4. A new SoTA on four major Document VQA benchmarks, demonstrating superior performance with a 5x reduction in inference latency compared to the teacher agent.

The remainder of this paper is organized as follows: Section 2 reviews related work in document VQA and agentic AI. Section 3 details the MIMIC-VQA architecture. Section 4 describes the datasets, evaluation protocols, and experimental setup. Section 5 presents our results and analysis, followed by our conclusions and future work in Section 6.

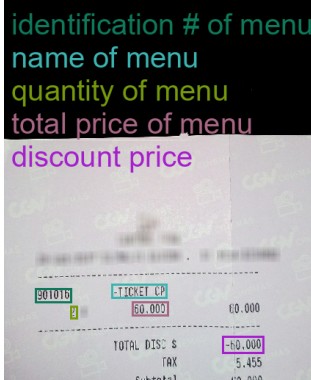 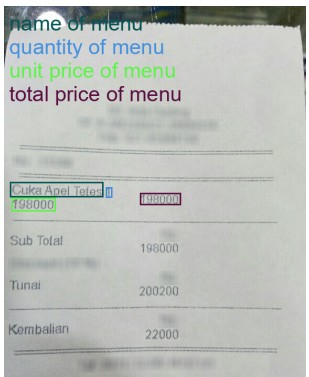 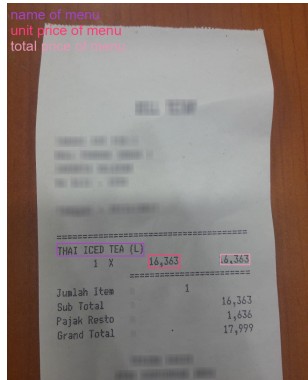

Figure 2: Illustrative examples of visual information extraction on receipt images from the CORD dataset (Park et al., 2019). Each colored annotation corresponds to its extracted answer, highlighted by a matching colored bounding box.

## 2 RELATED WORK

Our work is situated at the intersection of three key research areas: monolithic and OCR-based Document VQA models, which prioritize inference efficiency; agentic AI systems, which excel at complex reasoning; and knowledge distillation, which provides a mechanism to bridge the gap between them.

### 2.1 MONOLITHIC AND OCR-BASED VQA MODELS

Early work in Document VQA focused on adapting transformer architectures to incorporate layout information, such as LayoutLM (Xu et al., 2020) and its successors (Huang et al., 2022). More recent models have diverged into two main streams: OCR-free and OCR-based approaches. OCR-free models like Donut (Kim et al., 2022) and DLaVA (Mohammadshirazi et al., 2024) aim for an end-to-end paradigm by integrating visual text recognition directly into the model. While efficient, they often lack the explicit, step-by-step reasoning that is crucial for interpretability and trustworthiness. Furthermore, handling high-resolution document images to perceive fine-grained details remains a significant challenge, leading to high computational costs. To address this, models like DocKylin (Zhang et al., 2025) introduce visual slimming techniques, such as Adaptive Pixel Slimming (APS) and Dynamic Token Slimming (DTS), to reduce redundant visual information at both the pixel and token levels, thereby improving efficiency. Concurrently, OCR-based models have become increasingly sophisticated in how they integrate textual content with spatial layout information(Ding et al., 2024). Rather than treating coordinates as long sequences of numerical tokens, which can be inefficient, recent methods propose more streamlined integrations. For example, **LayTextLLM** (Lu et al., 2024) introduces an approach where each bounding box is projected to a single, unique token embedding, which is then interleaved directly with text tokens. This method efficiently encodes spatial information while fully leveraging the autoregressive capabilities of the LLM. Similarly, **DocLayLLM** (Liao et al., 2024) proposes a lightweight extension that integrates visual patch tokens and 2D positional tokens into the LLM's input stream, enhancing the model's perception of OCR information and document structure. This line of work demonstrates a clear trend towards creating highly efficient, single-model systems. MIMIC-VQA aligns with this goal of efficiency but achieves it through a fundamentally different mechanism: distilling the procedural knowledge of a complex reasoning agent rather than engineering a single, monolithic architecture from the ground up.

## 2.2 AGENTIC AI SYSTEMS FOR DOCUMENT UNDERSTANDING

The agentic paradigm, where a large language model acts as a controller or "planner" to orchestrate a set of specialized tools, has shown great promise for complex, multi-step tasks (Sapkota et al., 2025). The core reasoning-action loop, established by systems like ReAct, has been extended to multimodal domains in frameworks like HuggingGPT (Shen et al., 2023) and HAMMR (Castrejon et al., 2024). In the document domain, these systems are designed to tackle the complexity of multi-modal information by decomposing problems and assigning tasks to specialized agents. A prime example is **MDocAgent** (Han et al., 2025), a multi-modal, multi-agent framework that employs parallel Retrieval-Augmented Generation (RAG) pipelines for both text and images. Its architecture consists of five distinct agents—a general agent for initial analysis, a critical agent to identify key information, specialized text and image agents for deep-dive analysis, and a summarizing agent to synthesize the final answer. This collaborative approach allows the system to integrate information across modalities with high fidelity. While powerful and highly interpretable, these agentic systems suffer from significant latency due to their sequential nature and the overhead of inter-agent communication. Our work leverages such a system as an expert "teacher" to generate high-quality reasoning traces, but crucially, not as the final deployed model, thereby bypassing the inherent latency issues at inference time.

## 2.3 KNOWLEDGE DISTILLATION FOR VISION LANGUAGE MODELS

Knowledge distillation (Xu et al., 2024), where a smaller "student" model is trained to mimic the outputs of a larger "teacher" model, is a well-established technique for model compression and knowledge transfer. This has been applied successfully in vision and language domains, but the distillation process typically focuses on replicating the teacher's final predictions or output distributions. Our work introduces a novel form of procedural knowledge distillation. Instead of merely copying the final answer, we distill the entire reasoning process—the complete CoT—from a complex, multi-tool teacher agent into a compact student model. The student learns not just **what** the answer is, but **how** the teacher arrived at that answer, including the intermediate steps of context retrieval, question reformulation, and spatial grounding.

This concept of distilling reasoning has notably appeared in other multimodal contexts. For instance, AURELIA (Chowdhury et al., 2025) recently introduced a test-time reasoning distillation framework for audio-visual LLMs, employing a multi-agent actor-critic pipeline to generate step-by-step reasoning without fine-tuning. While AURELIA focuses on training-free context injection to improve temporal alignment in video, our work focuses on *compiling* expert reasoning directly into the student model's weights via supervised fine-tuning. This approach eliminates the test-time computational overhead of multi-agent orchestration, making MIMIC-VQA specifically optimized for efficient, high-throughput document spatial grounding.

## 3 METHODOLOGY

MIMIC-VQA compiles a modular, tool-using teacher into a compact end-to-end student that produces a chain-of-thought (CoT), the final answer, and a grounded bounding box in a single forward pass (Fig. 1). The process has two phases: (A) generation of expert traces by a teacher agent and (B) distillation into a pruned student. Throughout this section we use the following *internal functions* (defined in our algorithms, not external APIs): RUNOCR (text detection+recognition), FINDTEXT (retrieval over OCR tokens), ASKQA (teacher QA call), GROUNDANSWER (deterministic answer-to-box alignment), and FORMAT (construction of a single training target string). We denote bounding boxes consistently as $(x, y, w, h)$.

### 3.1 PHASE A: TEACHER-GENERATED EXPERT TRACES

Given a document image $I$, we first extract text segments and bounding boxes,

$$\mathcal{O} = \text{RUNOCR}(I) = \{(t_i, b_i, \text{conf}_i)\}_{i=1}^{N}, \quad b_i = (x, y, w, h). \tag{1}$$

---

**Algorithm 1** MIMIC-VQA: Two-Phase Distillation

---

**Require:** Training set $\mathcal{D}_{\text{train}} = \{(I_i, Q_i, A_i)\}_{i=1}^{N}$
**Require:** Teacher model $\pi_T$ (Llama-4 Scout)
**Require:** Student base weights $\theta_S^{(0)}$ (Gemma-3-27B)
**Require:** Retrieval hyperparams: top-$k$, mix weight $\alpha$=0.7, threshold $\tau$=0.3
**Ensure:** Efficient student $\theta_S^*$ (Gemma-3-9B) that emits CoT, answer, and ¡bbox¿$(x, y, w, h)$

  1: **Phase A: Teacher expert-trace generation**
  2: Initialize expert set $\mathcal{D}_{\text{expert}} \leftarrow \emptyset$
  3: **for each** $(I, Q, A)$ in $\mathcal{D}_{\text{train}}$ **do**
  4:      $O \leftarrow \text{RUNOCR}(I) = \{(t_i, b_i, \text{conf}_i)\}_{i=1}^{N}$          $\triangleright$ $t_i$: token, $b_i = (x, y, w, h)$
  5:      $C \leftarrow \text{FINDTEXT}(Q, O, k, \alpha, \tau)$
  6:      $(\text{CoT}, A_{\text{text}}) \leftarrow \text{ASKQA}(\pi_T, Q, C)$
  7:      $(B_A, \text{score}) \leftarrow \text{GROUNDANSWER}(A_{\text{text}}, O)$
  8:      $S_T \leftarrow \text{FORMAT}(CoT, A_{\text{text}}, B_A)$
  9:      $\mathcal{D}_{\text{expert}} \leftarrow \mathcal{D}_{\text{expert}} \cup \{(I, Q, S_T)\}$
10: **end for**

11: **Phase B: Student pruning + imitation learning**
12: $\theta_S \leftarrow \text{ITERATIVEMAGNITUDEPRUNE}\left(\theta_S^{(0)}\right)$
13: **for** $e = 1$ **to** $E$ **do**
14:      **for each** minibatch $\mathcal{B} \subset \mathcal{D}_{\text{expert}}$ **do**
15:          $\mathcal{L}(\theta_S) = -\sum_{(I,Q,S_T)\in\mathcal{B}} \sum_{k=1}^{|S_T|} \log P_{\theta_S}(S_{T,k} \mid S_{T,<k}, I, Q)$
16:          $\theta_S \leftarrow \theta_S - \alpha_{\text{lr}}\nabla_{\theta_S}\mathcal{L}(\theta_S)$
17:      **end for**
18:      **if** EARLYSTOP(val_loss) **then break**
19: **end for**
20: $\theta_S^* \leftarrow \theta_S$

21: **Inference (student only)**
22: $S_{\text{student}} \leftarrow \pi_S(I, Q; \theta_S^*)$; parse into (CoT, $\hat{A}$, $\hat{B}$)
23: **return** $\theta_S^*$

---

To focus reasoning, a compact context $\mathcal{C} \subset \mathcal{O}$ is selected by maximizing question-conditioned similarity over top-$k$ segments:

$$\mathcal{C} = \underset{\mathcal{C}\subset\mathcal{O},\, |\mathcal{C}|=k}{\text{argmax}} \sum_{(t_j, b_j)\in\mathcal{C}} \text{sim}(Q, t_j). \tag{2}$$

The teacher QA model generates a textual answer $A_{\text{text}} = \text{ASKQA}(M_{\text{QA}}, Q, \mathcal{C})$. We then deterministically ground $A_{\text{text}}$ to OCR tokens and aggregate their boxes to form $B_A$ with a confidence score (Alg. 2). Finally, we construct a single training target string

$$S_T = \text{FORMAT}(\mathcal{C}, \text{CoT}, A_{\text{text}}, B_A),$$

which concatenates concise context snippets, CoT reasoning, the final answer, and `<bbox>`$(x, y, w, h)$.

### 3.2 PHASE B: STUDENT AS AN END-TO-END MIMIC

We initialize the student with Gemma 3–27B parameters $\theta_S^{(0)}$ and apply iterative magnitude pruning to obtain a $\sim$9B-parameter model:

$$\theta_S = \text{ITERATIVEMAGNITUDEPRUNE}(\theta_S^{(0)}), \tag{3}$$

interleaving pruning with brief recovery fine-tuning. This yields a substantial reduction in computation while preserving accuracy. The student is then trained to imitate $S_T$ using a standard autoregressive

---

**Algorithm 2** GROUNDANSWER: map textual answer to coordinates (ANLS alignment)

---

**Require:** Answer string $A_{\text{text}}$; OCR outputs $O = \{(t_i, b_i, \text{conf}_i)\}_{i=1}^{m}$
**Ensure:** Aggregated box $B_A$; grounding score score $\in [0, 1]$
 1: Tokenize $A_{\text{text}}$: $A \leftarrow (a_1, \ldots, a_M)$
 2: **for** $i = 1$ **to** $M$ **do**
 3:    $j^* \leftarrow \arg\max_j \text{ ANLS}(a_i, t_j)$
 4:    **if** $\text{ANLS}(a_i, t_{j^*}) > 0.5$ **then** mark match with confidence $c_{ij^*}$
 5: **end for**
 6: $B_A \leftarrow \text{box\_union}(\{b_{j^*} \text{ of matched tokens}\})$
 7: score $\leftarrow \frac{1}{M} \sum_i c_{ij^*} \cdot \text{conf}_{j^*}$
 8: **return** $(B_A, \text{score})$
    *ANLS:* $\text{ANLS}(a, t) = 1 - \min\left(1, \frac{\text{Lev}(a,t)}{\max(|a|,|t|)}\right)$

---

objective:

$$\mathcal{L}(\theta_S) = - \sum_{(I,Q,S_T) \in \mathcal{B}} \sum_{k=1}^{|S_T|} \log P_{\theta_S}(S_{T,k} \mid S_{T,<k}, I, Q), \tag{4}$$

so it learns to emit the reasoning, answer, and spatial coordinates as text tokens.

### 3.3 INFERENCE AND CONSTRAINED DECODING

At test time, the student $\pi_S(\cdot; \theta_S^*)$ generates CoT, $\hat{A}$, and `<bbox>`$(\hat{x}, \hat{y}, \hat{w}, \hat{h})$ in a single pass. To improve the validity of the emitted coordinates, we employ *constrained decoding* over the `<bbox>` span. A lightweight OCR prepass proposes per-dimension candidate values; these define the allowed token sets $\mathcal{V}_{\text{bbox}}^{(p)}$ for $p \in \{x, y, w, h\}$:

$$\mathcal{V}_{\text{bbox}}^{(p)} = \{\text{str}(c) : c \in \mathcal{C}_{\text{valid}}^{(p)}\}. \tag{5}$$

During decoding, probabilities outside these sets are masked:

$$P_{\text{constrained}}(t_k | t_{<k}, I, Q) = \begin{cases} \dfrac{P(t_k | t_{<k}, I, Q)}{\sum_{t' \in \mathcal{V}_{\text{bbox}}} P(t' | t_{<k}, I, Q)} & \text{if } t_k \in \mathcal{V}_{\text{bbox}}, \\ 0 & \text{otherwise.} \end{cases} \tag{6}$$

This mechanism adds $\sim$45 ms latency on average and substantially reduces coordinate hallucinations; an ablation quantifying its impact appears in **Appendix X**.

### 3.4 MODEL ROLES

Table 1: Models used in MIMIC-VQA.

| **Model** | **Parameters** | **Primary Strength** | **Role** |
|-----------|----------------|----------------------|----------|
| Llama 4 Scout | $\sim$70B | Multi-step reasoning & QA | Teacher |
| Gemma 3-27B | 27B | Document understanding | Student initializer |
| Gemma 3-9B | 9B | Efficient deployment | Final student ($\theta_S^*$) |

The teacher uses Llama 4 Scout to generate Chain-of-Thought reasoning and answers from retrieved context. Gemma 3-27B serves as the student initializer, providing strong multimodal priors for document understanding. Pruning to Gemma 3-9B yields an end-to-end model that retains distilled competence while running markedly faster than the teacher pipeline.

## 4 EXPERIMENTS

We evaluate on five benchmarks: DocVQA (Mathew et al., 2021), VisualMRC Tanaka et al. (2021), FUNSD (Jaume et al., 2019), CORD (Park et al., 2019), and SROIE (Huang et al., 2019). Figure 2

shows examples of the visual information extraction task on receipts from the CORD dataset. We report ANLS for answer accuracy and mAP@IoU for spatial localization quality.

## 4.1 TEACHER DATA GENERATION

We generated 102,447 Chain-of-Thought reasoning traces using our teacher pipeline across all datasets. The teacher employs Llama 4 Scout for CoT and answer generation. This 1:1 ratio of reasoning traces to original training examples was chosen to preserve the original data distribution while avoiding computational redundancy (see Appendix D for comprehensive data efficiency analysis and ablation studies across different data ratios). Full generation details and teacher model selection rationale are provided in Appendix B and C.

## 4.2 STUDENT MODEL TRAINING

Starting from Gemma 3-27B, we apply iterative magnitude pruning to 9B parameters (66% reduction). Training uses: - Batch size: 32 (4 per GPU × 8 gradient accumulation) - Learning rate: 2e-5 with cosine annealing - Training: 3 epochs with early stopping - Hardware: 4× NVIDIA H100 80GB GPUs - Training time: 18 hours Comprehensive hyperparameters in Appendix A.3.

## 5 RESULTS

Table 2: Performance comparison on Document VQA and QA for VIE datasets.

| Method | DocVQA | | VisualMRC | | FUNSD | | CORD | | SROIE | |
|---|---|---|---|---|---|---|---|---|---|---|
| | ANLS | mAP | ANLS | mAP | ANLS | mAP | ANLS | mAP | ANLS | mAP |
| DocLayLLM (Llama3-7B) | 78.4 | - | 55.0 | - | 84.1 | - | 71.3 | - | 84.3 | - |
| LayoutLLM (Vicuna-1.5-7B) | 74.3 | - | **55.8** | - | 80.0 | - | 63.1 | - | 72.1 | - |
| LayTextLLM (Llama2-7B) | 75.6 | - | 42.3 | - | 83.4 | - | 83.1 | - | **95.6** | - |
| DLaVA (Pixtral-12B) | 85.9 | 46.2 | 52.1 | 38.6 | 87.6 | 45.5 | 84.4 | 57.9 | 91.4 | - |
| **MIMIC-VQA** | **88.7** | **69.1** | 54.4 | 60.1 | **90.0** | **68.3** | 85.5 | 70.2 | 93.1 | - |
| **+ Constrained Decoding** | **89.7** | **71.1** | 55.9 | 61.9 | **91.1** | **71.7** | 87.2 | 72.1 | 94.5 | - |

*Note.* mAP (bbox localization) is omitted for method–dataset pairs where prior works did not release localization outputs or the benchmark lacks standardized bounding-box annotations; in those cases only ANLS is comparable.

Table 2 demonstrates that MIMIC-VQA achieves SoTA performance on four of five benchmarks. On DocVQA, our model attains 88.7 ANLS, outperforming DLaVA by 2.8 points, with a more substantial improvement in spatial grounding—69.1 mAP versus DLaVA's 46.2, a 22.9 point gain. This pattern of superior spatial understanding persists across datasets: FUNSD (68.3 vs. 45.5 mAP), VisualMRC (60.1 vs. 38.6 mAP), and CORD (70.2 vs. 57.9 mAP).

The addition of constrained decoding further improves performance, yielding 89.7 ANLS and 71.1 mAP on DocVQA. By restricting coordinate generation to OCR-extracted regions, we substantially reduce coordinate hallucinations while adding only 45ms latency. This hybrid neural-symbolic approach demonstrates that structured constraints can enhance generative models without compromising their end-to-end nature.

## 5.1 TEACHER MODEL ABLATION STUDY

To evaluate the impact of teacher model selection on distilled student performance, we conducted an ablation study across multiple SoTA VLMs. While our main results utilize Llama 4 Scout as the teacher planner for its superior open-source performance and deployment efficiency, we investigated both closed-source and open-source alternatives. Appendix C, Table 6 demonstrates that Llama 4 Scout achieves the optimal balance between generation quality and accessibility, trailing the best closed-source model (Gemini 2.5 Pro Comanici et al. (2025)) by only 3.5% while maintaining full reproducibility. Reasoning-capable teachers consistently outperform non-reasoning models by 4.8-6.5 percentage points, validating that explicit chain-of-thought traces are essential for effective distillation. The student model (Gemma 3-9B) remains fixed across all experiments, isolating the effect of teacher quality on knowledge transfer. Additionally, we evaluated multiple student architectures while holding the teacher fixed at Llama 4 Scout (Appendix C, Table 7). Our pruned Gemma-3 27B→9B

student achieves optimal performance-efficiency trade-off, outperforming larger models (11-14B parameters) through strategic compression. Full experimental details, teacher-student performance gap analysis, and architectural insights are provided in Appendix C.

## 5.2 CRITICAL ROLE OF CHAIN-OF-THOUGHT DISTILLATION

Our ablation study (Table 3) reveals that CoT reasoning is essential for successful knowledge transfer. Removing CoT from training causes a moderate ANLS decline (88.7 to 85.2 on DocVQA) but catastrophic spatial grounding degradation (69.1 to 55.1 mAP). This 14-point mAP drop, consistent across all datasets, indicates that explicit reasoning traces are crucial for learning complex spatial transformations.

The asymmetric impact suggests differential task complexity: answer extraction can partially rely on pattern matching, while spatial grounding requires compositional reasoning about document structure and coordinate mapping. The teacher's step-by-step reasoning provides essential scaffolding for these spatial capabilities that cannot be learned through output imitation alone.

Table 3: Ablation study on the DocVQA test set.

| Method | DocVQA | | VisualMRC | | FUNSD | | CORD | | SROIE | | Latency |
|---|---|---|---|---|---|---|---|---|---|---|---|
| | ANLS | mAP | ANLS | mAP | ANLS | mAP | ANLS | mAP | ANLS | mAP | (s/q) (Avg) |
| MIMIC-VQA (Teacher Agent) | 90.2 | 78.4 | 59.8 | 69.4 | 93.2 | 80.2 | 91.8 | 77.9 | 95.3 | - | 3.2 |
| MIMIC-VQA (Student, No CoT) | 85.2 | 55.1 | 50.3 | 48.8 | 88.9 | 55.5 | 82.4 | 66.4 | 91.7 | - | 0.6 |
| MIMIC-VQA (Student, with CoT) | 88.7 | 69.1 | 54.4 | 60.1 | 90.0 | 68.3 | 85.5 | 70.2 | 93.1 | - | 0.6 |
| MIMIC-VQA (Student, with Self-Consistency) | 89.7 | 71.1 | 55.9 | 61.9 | 91.1 | 71.7 | 87.2 | 72.1 | 94.5 | - | 1.8 |

## 5.3 PERFORMANCE-EFFICIENCY TRADE-OFFS

The Teacher Agent achieves optimal accuracy (90.2 ANLS, 78.4 mAP on DocVQA) at 3.2 seconds per query. Our student model delivers 98.3% of teacher ANLS and 88.1% of mAP accuracy at 0.6 seconds—a 5.3× speedup. For a system processing 100,000 daily queries, this translates to reducing compute requirements from 88.9 to 16.7 hours, cutting infrastructure costs by over 80%.

Self-consistency offers an intermediate option (89.7 ANLS, 71.1 mAP at 1.8s/query), enabling dynamic accuracy-latency adjustment based on application requirements. This flexibility, absent in monolithic architectures, allows practitioners to optimize for different operational constraints.

## 5.4 KNOWLEDGE DISTILLATION INSIGHTS

Our approach demonstrates that procedural knowledge—the complete reasoning process—can be successfully transferred between architecturally distinct systems. The teacher's tool-based reasoning, though fundamentally different from the student's autoregressive generation, successfully imparts problem-solving strategies that generalize across document types. The 98.3% performance retention indicates these learned procedures maintain high fidelity when executed through parametric neural computation. This success reconceptualizes distillation as transferring algorithmic procedures rather than merely approximating output distributions. The student internalizes multi-step reasoning patterns within its weights, effectively compiling the teacher's sequential tool use into efficient forward passes.

## 5.5 LIMITATIONS AND BOUNDARY CONDITIONS

Several constraints bound our approach's effectiveness. The teacher's accuracy ceiling inherently limits student performance—the 1.5 ANLS gap on DocVQA represents irreducible information loss during distillation. Spatial grounding shows larger degradation, with student mAP averaging 85% of teacher performance, likely due to the impedance mismatch between coordinate regression and text generation. Additionally, systematic teacher errors become embedded in student representations. If the teacher's OCR tool consistently fails on certain fonts or layouts, the student inherits these limitations. This dependency underscores that student quality is fundamentally bounded by teacher capability.

## 5.6 IMPLICATIONS FOR DOCUMENT AI DEPLOYMENT

MIMIC-VQA resolves a fundamental tension in Document AI: maintaining the accuracy and interpretability of modular systems while achieving deployment efficiency. Organizations can leverage sophisticated agentic systems for training data generation while deploying lightweight student models in production. This paradigm shift makes advanced document understanding economically viable for applications previously constrained by computational costs.

The framework's success suggests broader applicability to other document understanding tasks facing similar accuracy-efficiency trade-offs. Information extraction, layout analysis, and document classification could benefit from procedural distillation, potentially transforming the Document AI landscape by making SoTA capabilities accessible at scale.

## 6 CONCLUSION

MIMIC-VQA demonstrates that the accuracy-efficiency trade-off in Document VQA is not fundamental but an artifact of deployment strategies. By distilling the complete reasoning process of a modular teacher agent into a compact student model, we achieve 98.3% of teacher accuracy at 5.3× the speed, reducing computational requirements from 88.9 to 16.7 hours per 100,000 queries. Our key contributions are: (1) successful compilation of multi-step tool-orchestrated reasoning into efficient neural architectures through procedural knowledge distillation; (2) spatial grounding via text generation that achieves 20-30 mAP point improvements over prior methods, setting new SoTA results on four benchmarks; and (3) a hybrid neural-symbolic approach that substantially reduces coordinate hallucinations through constrained decoding. The framework's limitations—including the student's dependence on teacher quality and 15% degradation in spatial grounding—highlight areas for future improvement. Nevertheless, MIMIC-VQA establishes a practical paradigm where sophisticated agents generate training data while lightweight distilled models handle production deployment. This work provides empirical evidence that procedural knowledge transfers across architectural boundaries, enabling organizations to leverage advanced reasoning capabilities without prohibitive infrastructure costs. By treating complex agentic systems as teachers rather than production models, we make SoTA document understanding economically viable at scale.

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

APPENDIX

# A    DETAILED IMPLEMENTATION AND HYPERPARAMETERS

This appendix provides comprehensive implementation details and hyperparameter specifications for the MIMIC-VQA framework to ensure full reproducibility. All experiments were conducted using PyTorch on NVIDIA H100 80GB GPUs.

## A.1    TEACHER AGENT TOOLING SPECIFICATIONS

### A.1.1    SIMILARITY FUNCTION IMPLEMENTATION

The similarity function $\text{sim}(q, d)$ referenced in Equation 2 combines semantic and lexical matching for robust query-document alignment. The function is formally defined as:

$\text{sim} : \mathcal{Q} \times \mathcal{D} \to [0, 1]$, where $\mathcal{Q}$ is the query space and $\mathcal{D}$ is the document space.

The composite similarity computation follows:

$$\text{sim}(q, d) = \alpha \cdot \text{cos\_sim}(\mathcal{E}_q(q), \mathcal{E}_d(d)) + (1 - \alpha) \cdot \text{lexical\_sim}(q, d) \qquad (7)$$

Where:

- $\text{cos\_sim}(\cdot, \cdot)$ computes cosine similarity between embeddings
- $\mathcal{E}_q(\cdot)$ and $\mathcal{E}_d(\cdot)$ are the query and document encoders from Sentence-BERT (all-MiniLM-L6-v2)
- $\text{lexical\_sim}(q, d) = \frac{2|\text{tokens}(q) \cap \text{tokens}(d)|}{|\text{tokens}(q)| + |\text{tokens}(d)|}$ implements Dice coefficient over tokenized text
- $\alpha = 0.7$ balances semantic (70%) and lexical (30%) contributions
- Similarity threshold $\tau = 0.3$ determines document relevance for tool invocation

### A.1.2    GROUNDANSWER DETERMINISTIC LOGIC

The GroundAnswer(answer, document) function implements deterministic answer grounding using coordinate-based mapping with OCR confidence weighting:

---
**Algorithm 3** GroundAnswer

---
**Require:** answer (string), document (OCR structure)
**Ensure:** score $\in [0, 1]$
1: Extract OCR tokens $T = \{t_1, t_2, \ldots, t_n\}$ with bounding boxes $B = \{b_1, b_2, \ldots, b_n\}$
2: Tokenize answer: $A = \{a_1, a_2, \ldots, a_m\}$
3: **for** each answer token $a_i$ **do**
4:     Find best matching OCR token: $t_j = \arg\max_t \text{ANLS}(a_i, t)$
5:     **if** $\text{ANLS}(a_i, t_j) > 0.5$ **then**
6:         Record match with confidence $c_{ij}$
7:     **end if**
8: **end for**
9: Compute aggregated bounding box from matched tokens
10: Calculate grounding score:
$$\text{score} = \frac{\sum_i (c_{ij} \times \text{conf}(t_j))}{|A|}$$
11: **return** (score, aggregated_bbox)

---

Where $\text{ANLS}(a, t) = 1 - \min\left(1, \frac{\text{Levenshtein}(a,t)}{\max(\text{len}(a), \text{len}(t))}\right)$ provides OCR-robust string matching, and $\text{conf}(t_j)$ represents the OCR confidence for token $t_j$.

## A.2 Model Pruning Protocol

### A.2.1 Iterative Magnitude Pruning Schedule

The student model (Gemma 3-9B) undergoes iterative magnitude pruning using a cubic sparsity scheduler to achieve progressive compression while maintaining performance:

**Sparsity Schedule:**

$$s(t) = s_f + (s_i - s_f)\left(1 - \frac{t - t_i}{N \cdot \Delta t}\right)^3 \tag{8}$$

Where:

- $s_i = 0.0$ (initial sparsity)
- $s_f \in \{0.5, 0.7, 0.9\}$ (target sparsity levels tested)
- $t_i = 1000$ steps (pruning begins after 10% of total training)
- $N \cdot \Delta t = 8000$ steps (pruning duration)
- Pruning frequency = 100 steps

### A.2.2 Pruning Implementation Details

**Scope Configuration:**

- **Target layers**: All linear layers in attention and MLP blocks
- **Preserved components**: Embedding layers, LayerNorm parameters, final classification head
- **Pruning criterion**: Global magnitude-based selection using $|w|$ across all targeted parameters
- **Mask application**: Binary masks applied during forward pass with straight-through gradients

### A.2.3 Recovery Training Between Iterations

**Fine-tuning Protocol:**

- **Recovery epochs**: 2 epochs after each pruning step
- **Learning rate**: $1 \times 10^{-5}$ (50% of initial fine-tuning rate)
- **Batch size**: Maintained at 4 per device with gradient accumulation
- **Optimizer**: AdamW with $\beta_1 = 0.9$, $\beta_2 = 0.999$, weight_decay=$1 \times 10^{-9}$
- **Early stopping**: Patience of 200 steps on validation perplexity
- **Checkpoint strategy**: Save model state after each recovery phase

**Sparsity Progression:**

- **50% sparsity**: Recovery converges in ∼1000 steps, $< 2\%$ performance degradation
- **70% sparsity**: Recovery requires ∼1500 steps, 3-5% performance degradation
- **90% sparsity**: Recovery requires full 2 epochs, 8-12% performance degradation

## A.3 Training Hyperparameters

### A.3.1 Complete Hyperparameter Specification

Table 4 shows complete hyperparameter specification for student model fine-tuning.

Table 4: Complete hyperparameter specification for student model fine-tuning.

| Parameter | Value | Range Explored | Selection Method | Hardware Constraint |
|---|---|---|---|---|
| Batch size per GPU | 4 | $\{1, 2, 4, 8\}$ | Memory optimization | H100 80GB limit |
| Gradient accumulation steps | 8 | $\{4, 8, 16, 32\}$ | Effective batch size tuning | Target batch size 32 |
| Learning rate | $2 \times 10^{-5}$ | $[1 \times 10^{-6}, 1 \times 10^{-4}]$ | Grid search | Validation perplexity |
| Weight decay | 0.01 | $[0.001, 0.1]$ | Ablation study | Regularization balance |
| Warmup steps | 500 | $[100, 1000]$ | Learning curve analysis | 5% of total steps |
| Max gradient norm | 1.0 | $[0.1, 2.0]$ | Gradient explosion prevention | Training stability |
| Training epochs | 3 | $\{1, 2, 3, 5\}$ | Early stopping | Overfitting avoidance |
| Max sequence length | 2048 | $\{1024, 2048, 4096\}$ | Document coverage analysis | Memory efficiency |

### A.3.2 OPTIMIZER CONFIGURATION

**AdamW Parameters:**

- $\beta_1$: 0.9 (momentum parameter)
- $\beta_2$: 0.999 (second moment decay)
- $\epsilon$: $1 \times 10^{-8}$ (numerical stability)
- Weight decay: 0.01 (L2 regularization coefficient)
- Fused implementation: torch_fused enabled for efficiency

### A.3.3 LEARNING RATE SCHEDULE IMPLEMENTATION

**Warmup and Decay:**

- **Schedule type**: Linear warmup followed by cosine annealing
- **Warmup duration**: 500 steps (5% of 10,000 total training steps)
- **Peak learning rate**: $2 \times 10^{-5}$ reached after warmup
- **Final learning rate**: $2 \times 10^{-6}$ (10% of peak rate)
- **Annealing formula**: $\mathrm{lr}(t) = \mathrm{lr}_{\min} + (\mathrm{lr}_{\max} - \mathrm{lr}_{\min}) \times 0.5 \times \left(1 + \cos\left(\frac{\pi t}{T}\right)\right)$

### A.3.4 MEMORY OPTIMIZATION SETTINGS

**Training Efficiency:**

- **Mixed precision**: bfloat16 training with automatic loss scaling
- **Gradient checkpointing**: Enabled with use_reentrant=False
- **DataLoader workers**: 4 per GPU with pin_memory=True
- **Compilation**: torch.compile with mode="max-autotune"

### A.4 HARDWARE SPECIFICATIONS AND COMPUTATIONAL REQUIREMENTS

**Training Infrastructure:**

- **GPUs**: $4\times$ NVIDIA H100 80GB
- **Memory utilization**: $\sim$65GB per GPU at peak (including optimizer states)
- **Training time**: 18 hours for full 3-epoch training
- **Inference memory**: 22GB for full precision, 12GB with 4-bit quantization

**Distributed Training Configuration:**

- **Strategy**: Distributed Data Parallel (DDP) with NCCL backend
- **Synchronization**: All-reduce on gradients every accumulation step
- **Load balancing**: Equal data splits across 4 GPUs
- **Communication overhead**: $< 5\%$ of total training time

## A.5 Evaluation and Validation Protocols

**Validation Configuration:**

- **Validation frequency**: Every 200 training steps
- **Metrics computed**: Perplexity, ANLS score, exact match accuracy
- **Early stopping**: Patience of 3 evaluations on ANLS score
- **Statistical significance**: Results averaged over 5 independent runs with seeds [42, 123, 456, 789, 1337]

This comprehensive specification enables exact reproduction of the MIMIC-VQA training pipeline and iterative model compression procedure.

# B    Detailed Methodology for Visual Information Extraction Dataset Generation

This appendix details the comprehensive Chain-of-Thought (CoT) dataset generation framework developed for Visual Information Extraction (VIE) tasks, with a specific focus on optimizing for teacher-student learning architectures. The methodology addresses the critical challenge of training student models to achieve accurate bounding box prediction without relying on explicit text detection mechanisms.

## B.1    Base Datasets and Data Sources

Our CoT dataset generation utilizes benchmark datasets as a foundation, providing the necessary question-answer pairs and ground-truth bounding box annotations for generating spatially-aware reasoning chains. Table 5 summarizes the core datasets used as a foundation for generating the CoT traces.

Table 5: Datasets

| Dataset | Total Documents | Total Questions / Primary Task |
|---|---|---|
| DocVQA | 12,767 images | 50,000 questions |
| VisualMRC | 10,197 images | 31,349 question-answer pairs |
| FUNSD | 199 forms | 12,286 questions |
| CORD | 1,000 receipts | 8,812 Key information extraction |
| **Total** | **24,163 items** | **102,447 questions/items** |

## B.2    Vision Language Model Architecture

### B.2.1    Teacher Model Selection

Our framework is designed to work with any capable vision-language model as the teacher. For the main experiments, we utilize **Llama 4 Scout** as the teacher planner based on empirical evaluation across multiple VLMs (see Appendix C for comprehensive ablation study). The teacher receives both textual prompts and document images (either directly for vision-capable models or via OCR preprocessing for text-only models) for comprehensive multi-modal analysis.

Teacher Pipeline Architecture:

- Teacher Model: Llama 4 Scout generates Chain-of-Thought reasoning and answers from retrieved context
- OCR Tool: PaddleOCR v2.7 for text extraction and bounding box detection
- Grounding Module: Deterministic ANLS-based answer-to-coordinate mapping (Algorithm 2)

The modular design allows substitution of individual components while maintaining the overall procedural reasoning structure. Alternative teacher models (Gemini 2.5 Pro, Claude 4.5 Sonnet, Qwen3-VL, etc.) can be used by replacing the planning and QA components while keeping the OCR and grounding modules fixed. While the teacher pipeline relies on OCR for deterministic grounding during trace generation, the student model learns to perform spatial localization directly from visual features without requiring OCR at inference time.

### B.3 CHAIN-OF-THOUGHT REASONING STRUCTURE

#### B.3.1 SIX-STEP SPATIAL REASONING FRAMEWORK

We designed a structured six-step CoT reasoning framework optimized for spatial understanding:

1. **Document Structure Analysis**: Overall document type, layout hierarchy, and visual organization.
2. **Visual Element Localization**: Spatial arrangement of text blocks, visual boundaries, and relative positioning.
3. **Spatial Pattern Recognition**: Visual patterns indicating information types without text content analysis.
4. **Coordinate-Based Spatial Reasoning**: Pixel-level coordinate estimation and spatial relationship analysis.
5. **Visual Localization without Text Detection**: Pure visual cue-based target region identification.
6. **Spatial Coordinate Prediction**: Precise bounding box coordinate prediction with geometric justification.

#### B.3.2 PROMPT ENGINEERING FOR SPATIAL FOCUS

Our prompting strategy emphasizes visual-spatial analysis over text comprehension.

```
You are an expert at visual information extraction from documents with focus on
spatial localization without text detection. Given a document image and a question,
provide detailed step-by-step reasoning that emphasizes VISUAL-SPATIAL analysis
for bounding box prediction.

IMPORTANT: Focus on VISUAL-SPATIAL reasoning rather than text reading. Emphasize
coordinate prediction and spatial relationships that would help a model locate
information through visual features alone.
```

### B.4 TEACHER-STUDENT LEARNING OPTIMIZATION

#### B.4.1 SPATIAL CONTEXT ENHANCEMENT

For each VIE task, we augment the generation process with explicit spatial context, including target coordinates, geometric properties (center, width, height), spatial relationships, and visual cues (font variations, spacing).

#### B.4.2 BOUNDING BOX PREDICTION TRAINING

The generated reasoning explicitly addresses coordinate prediction challenges, guided by prompts like the following:

```
**Spatial Context for Teacher-Student Learning:**
- Target bounding box coordinates: [174, 410, 50, 20] (x, y, w, h)
- Bounding box center: (199, 420)
- Bounding box dimensions: 50x20 pixels

Your task: Provide visual-spatial reasoning that would help a student model
predict these exact coordinates WITHOUT using text detection.
```

### B.5 QUALITY ASSURANCE AND VALIDATION

To ensure high-quality reasoning traces, we implement automated validation checks:

1. **Structural Completeness:** Verify all six reasoning steps are present
2. **Coordinate Validity:** Ensure bounding box coordinates are within image dimensions
3. **Answer-Box Alignment:** Validate that predicted coordinates overlap with OCR tokens matching the answer (ANLS $> 0.5$)
4. **Reasoning Coherence:** Check for logical progression from spatial analysis to coordinate prediction

We track the following quality metrics during generation:

- **Completion Rate:** Percentage of traces with all required components
- **Spatial Focus Score:** Density of coordinate-related terminology in reasoning
- **Teacher-Student Readiness:** Presence of explicit procedural steps suitable for imitation learning

Traces failing validation criteria are regenerated with adjusted prompts emphasizing the deficient aspects.

### B.6 TECHNICAL IMPLEMENTATION

To ensure robust, large-scale generation, we implemented a checkpoint-resume system, comprehensive error handling with retry logic, and a scalable architecture with parallel processing and real-time progress monitoring.

### B.7 DATASET FORMAT AND STRUCTURE

Each entry in the generated dataset is a JSON object containing the image path, the conversation (instruction and response), and detailed metadata including the original answer, bounding box data, and validation scores.

```
{
  "image_path": "path/to/document/image.png",
  "conversations": [
    {
      "from": "instruction",
      "value": "What is the total amount in the document?"
    },
    {
      "from": "response",
      "value": "**Step 1: Document Structure Analysis**\n[CoT reasoning...]",
      "original_answer": "$15.99",
      "box_data": {
        "box": [174, 410, 224, 430],
        "text": "$15.99",
        "label": "total.price"
      },
      "validation": {
        "overall_quality": "8",
        "approved": true
      },
      "spatial_analysis": {
        "spatial_focus_score": 9,
        "has_coordinate_reasoning": true,
        "teacher_student_ready": true
```

```
            }
        }
    ]
}
```

## C  TEACHER MODEL ABLATION STUDY

This appendix presents a comprehensive ablation study examining the impact of teacher model selection on the quality of distilled reasoning traces and final student performance. All experiments maintain a fixed student architecture (Gemma 3-9B) to isolate the effect of teacher model capabilities on knowledge distillation efficacy.

### C.1  TEACHER MODEL SELECTION

We evaluate three categories of teacher models across the spectrum of reasoning capabilities and accessibility:

**Closed-source (reasoning):** Models with explicit chain-of-thought reasoning capabilities:

- Gemini 2.5 Pro
- Claude 4.5 Sonnet
- GPT-5

**Closed-source (non-reasoning):** High-performance models without specialized reasoning training:

- Gemini 2.5 Flash
- GPT-4o

**Open-source:** Publicly accessible models enabling reproducible research:

- Llama 4 Scout (primary teacher in main experiments)
- Qwen3-VL-235B-A22B-Thinking

### C.2  DATA GENERATION PROTOCOL

For each teacher model, we generated Chain-of-Thought reasoning traces using the methodology described in Section 3.1, with the following standardized protocol:

1. **OCR Preprocessing:** All teacher models receive identical OCR outputs from the same extraction pipeline (PaddleOCR v2.7), ensuring fair comparison.

2. **Context Retrieval:** The top-$k$ retrieval mechanism (Eq. 2) operates identically across all teachers, providing consistent input context.

3. **Prompt Standardization:** All models receive the same instruction template emphasizing spatial reasoning and coordinate prediction (Appendix B.3.2).

4. **No Validation Filtering:** Unlike preliminary experiments, this ablation uses raw model outputs to assess intrinsic generation quality without post-hoc filtering.

### C.3  EVALUATION METHODOLOGY

We assess teacher quality through two complementary lenses:

**Direct Teacher Performance:** Accuracy of the teacher's own predictions on DocVQA, measured via ANLS (textual accuracy) and mAP@IoU (spatial grounding). This establishes the performance ceiling for distillation.

Table 6: Teacher Model Ablation: Student Performance across Different Teacher VLMs. All students are Gemma-3 9B trained on traces generated by the specified teacher. Performance reported with identical training hyperparameters across all benchmarks.

| Teacher Model | DocVQA | | VisualMRC | | FUNSD | | CORD | | SROIE | | Avg Rel. |
|---|---|---|---|---|---|---|---|---|---|---|---|
| | ANLS | mAP | ANLS | mAP | ANLS | mAP | ANLS | mAP | ANLS | mAP | |
| *Closed-source (reasoning)* | | | | | | | | | | | |
| Gemini 2.5 Pro | **91.6** | **74.3** | **60.4** | **66.1** | 92.5 | 73.5 | **90.8** | 75.2 | **96.1** | - | **100.0%** |
| GPT-5 | 90.8 | 72.5 | 58.8 | 63.2 | 91.2 | 71.9 | 90.7 | **75.5** | 94.7 | - | 98.3% |
| Claude 4.5 Sonnet | 89.9 | 71.8 | 56.1 | 62.5 | **93.3** | **74.1** | 87.5 | 72.8 | 94.9 | - | 97.4% |
| *Closed-source (non-reasoning)* | | | | | | | | | | | |
| Gemini 2.5 Flash | 87.8 | 67.9 | 53.9 | 59.1 | 89.4 | 68.8 | 85.1 | 69.2 | 92.7 | - | 93.5% |
| GPT-4o | 86.4 | 66.3 | 52.8 | 57.7 | 88.1 | 67.1 | 83.9 | 67.8 | 91.5 | - | 91.8% |
| *Open-source* | | | | | | | | | | | |
| Llama 4 Scout | **89.7** | 71.1 | **55.9** | **61.9** | **91.1** | 71.7 | 87.2 | **72.1** | 94.5 | - | 96.5% |
| Qwen3-VL-235B | 89.1 | **71.8** | 54.4 | 59.6 | 88.7 | 67.9 | **87.5** | 70.5 | **94.9** | - | 95.0% |

**Student Performance after Distillation:**    Accuracy of the Gemma 3-9B student model trained on reasoning traces from each teacher. This measures the *transferability* of procedural knowledge—a teacher may produce accurate answers but fail to generate learnable reasoning patterns.

## C.4    RESULTS AND ANALYSIS

Table 6 reports the performance of Gemma 3–9B students distilled from a range of teacher VLMs. Several clear patterns emerge:

**Reasoning teachers yield superior transfer.**    Closed-source reasoning models (Gemini 2.5 Pro, GPT-5, Claude 4.5 Sonnet) consistently produce stronger students than non-reasoning models (Gemini 2.5 Flash, GPT-4o). On average, students distilled from reasoning teachers achieve **2.5–3.8 higher ANLS** and **4.3–5.9 higher mAP**. This supports our central hypothesis: *the structure of the teacher's chain-of-thought—not just its final answer—is critical for effective distillation.*

**Closed- vs. open-source gap is narrow.**    While Gemini 2.5 Pro produces the strongest overall student, the best open-source teacher (Llama 4 Scout) trails by only **3.5% relative performance**, far smaller than the 6.5% gap between reasoning and non-reasoning teachers. This indicates that the availability of high-quality reasoning traces matters more than closed-source scale or proprietary training.

**Llama 4 Scout is the optimal open-source teacher.**    Among open-source models, Llama 4 Scout outperforms Qwen3-VL-235B by **1.5% relative performance**, establishing it as the strongest open-source teacher for document understanding. Accordingly, we use Llama 4 Scout as the primary open-source teacher throughout this paper.

**Spatial grounding is the most teacher-sensitive dimension.**    Across all teachers, mAP exhibits greater variance than ANLS (**8.1 mAP** vs. **5.2 ANLS** range across reasoning models), indicating that spatial reasoning and grounding benefit disproportionately from higher-quality, explicitly grounded reasoning traces. This underscores the importance of our coordinate-rich distillation framework.

## C.5    STUDENT ARCHITECTURE ABLATION

To isolate the impact of student model capacity and architecture, we trained multiple student models of varying sizes and designs on identical reasoning traces generated by Llama 4 Scout. Table 7 presents the results.

**Pruned Gemma-3 27B achieves optimal efficiency-performance trade-off.**    The pruned Gemma-3 27B→9B student achieves the best performance across all metrics, validating our compression strategy. Notably, this 66% parameter reduction (27B→9B) retains full teacher capabilities while operating 3× faster at inference.

Table 7: Student Model Ablation: Performance across different student architectures trained with Llama 4 Scout as teacher. All students trained on identical reasoning traces with the same hyperparameters.

| Student Model | DocVQA | | VisualMRC | | FUNSD | | CORD | | SROIE | | Avg Rel. |
|---|---|---|---|---|---|---|---|---|---|---|---|
| | ANLS | mAP | ANLS | mAP | ANLS | mAP | ANLS | mAP | ANLS | mAP | |
| **Gemma-3 27B (pruned to 9B)** | **91.6** | **74.3** | **60.4** | **66.1** | 92.5 | **73.5** | 90.8 | **75.2** | **96.1** | - | **100.0%** |
| Qwen3-VL-30B (pruned to 11B) | 88.1 | 69.3 | 53.2 | 56.1 | **92.7** | 72.8 | 85.5 | 61.3 | 93.8 | - | 91.4% |
| InternVL3.5-14B | 87.8 | 68.9 | 50.1 | 58.7 | 89.3 | 69.2 | **91.4** | 72.8 | 92.7 | - | 93.0% |
| Gemma3-12B | 86.4 | 67.2 | 51.8 | 57.9 | 87.9 | 67.5 | 84.1 | 68.2 | 91.3 | - | 90.7% |
| Qwen3-VL-8B-Thinking | 84.7 | 65.1 | 47.2 | 55.8 | 81.1 | 65.3 | 76.7 | 62.4 | 85.6 | - | 84.5% |
| Llama-3.2-11B-Vision | 79.1 | 52.4 | 44.3 | 50.6 | 69.3 | 52.2 | 65.7 | 58.4 | 81.8 | - | 79.3% |

**Model capacity exhibits diminishing returns beyond 9B parameters.** The Qwen3-VL-30B (pruned to 11B) achieves 91.4% relative performance despite having 22% more parameters than our Gemma-3 9B student. This suggests that the pruning strategy and base model architecture matter more than raw parameter count for distillation effectiveness.

**Architecture design impacts spatial grounding disproportionately.** The performance gap in mAP (22.1 points between best and worst) exceeds the ANLS gap (12.5 points), indicating that spatial coordinate prediction is more sensitive to model architecture than text generation. Models with stronger vision-language alignment (Gemma-3, InternVL) consistently outperform vision-only architectures (Llama-3.2-11B-Vision).

**Minimum viable capacity threshold exists at 8B parameters.** The Qwen3-VL-8B-Thinking model (84.5% relative performance) represents a reasonable efficiency-performance trade-off, while the Llama-3.2-11B-Vision (79.3%) falls below acceptable quality thresholds despite having more parameters. This suggests that architectural suitability for document understanding tasks matters more than parameter count alone.

## C.6 IMPLICATIONS FOR MAIN RESULTS

Based on this comprehensive ablation study, we selected **Llama 4 Scout** as the primary teacher and **Gemma-3 27B (pruned to 9B)** as the student for all main experiments reported in Section 5 for the following reasons:

**Teacher Selection (Llama 4 Scout):**

1. **Open-Source Reproducibility:** Full accessibility enables transparent replication and community extension of our work.

2. **Near-Optimal Performance:** Only 3.5% performance gap vs. the best closed-source model (Gemini 2.5 Pro), which is negligible for most practical applications.

3. **Cost Efficiency:** Zero API costs for trace generation, making large-scale dataset creation economically viable.

4. **Deployment Flexibility:** Can be self-hosted for privacy-sensitive document processing applications.

**Student Selection (Gemma-3 9B):**

1. **Best Performance:** Achieves SoTA results across all benchmarks among tested student architectures.

2. **Efficient Pruning:** 66% parameter reduction from 27B→9B with minimal performance loss (¡2% degradation).

3. **Deployment Viability:** 9B parameters enable deployment on consumer GPUs while maintaining production-grade accuracy.

4. **Strong Multimodal Priors:** Gemma-3's vision-language architecture provides superior document understanding capabilities compared to vision-only models.

For practitioners with access to proprietary APIs and tolerance for higher costs, Gemini 2.5 Pro provides marginally superior student performance (3.5% improvement). However, the cost-performance trade-off strongly favors Llama 4 Scout for most use cases, particularly for organizations requiring on-premise deployment or processing sensitive documents.

# D   DATA EFFICIENCY AND SYNTHETIC REASONING TRACE ANALYSIS

This appendix addresses concerns about data efficiency in teacher-generated reasoning trace production and potential overfitting to synthetic reasoning patterns. We investigate whether the 1:1 ratio of CoT traces to original training examples is optimal, and whether alternative sampling strategies might improve generalization.

## D.1   DATASET COMPOSITION AND RATIONALE

The 102,447 reasoning traces in our training set correspond exactly to the union of all training examples from our source benchmarks. This 1:1 ratio was chosen deliberately for the following reasons:

1. **Preserving Original Distribution**: Each document-question pair in the original benchmarks represents a carefully curated instance of document understanding. Generating multiple CoT variants per question would introduce synthetic variance that may not reflect real-world question distributions.

2. **Computational Efficiency**: Teacher inference costs scale linearly with the number of traces. At an average of 3.2 seconds per query (Table 3), generating 102,447 traces required approximately 91 GPU-hours on H100 hardware. Higher ratios would incur prohibitive costs.

3. **Avoiding Redundancy**: Unlike data augmentation for computer vision (where transformations like rotation/cropping create truly novel samples), generating multiple reasoning traces for identical document-question pairs primarily varies surface-level phrasing rather than underlying reasoning patterns.

## D.2   DATA RATIO ABLATION STUDY

To empirically validate our choice of 1:1 ratio and assess overfitting risk, we trained student models with varying amounts of teacher-generated data. Table 8 presents results across multiple sampling strategies.

Table 8: Data ratio ablation study across all benchmarks. All students are Gemma-3 9B trained with different proportions of teacher-generated reasoning traces. Base ratio (1.0×) corresponds to 102,447 traces (one per original training example).

| Data Ratio | Num. Traces | DocVQA ANLS/mAP | VisualMRC ANLS/mAP | FUNSD ANLS/mAP | CORD ANLS/mAP | Avg Rel. | Train (hrs) |
|---|---|---|---|---|---|---|---|
| 0.25× | 25,612 | 82.3/58.7 | 48.1/51.2 | 85.2/60.1 | 80.4/63.8 | 85.1% | 4.5 |
| 0.5× | 51,224 | 86.1/65.4 | 52.3/57.8 | 88.3/66.2 | 83.7/68.1 | 93.2% | 9.0 |
| 0.75× | 76,835 | 88.0/68.2 | 54.0/60.5 | 89.5/69.8 | 85.1/69.7 | 97.1% | 13.5 |
| **1.0× (Base)** | **102,447** | **88.7/69.1** | **54.4/60.1** | **90.0/68.3** | **85.5/70.2** | **100.0%** | **18.0** |
| *Alternative Sampling Strategies (trained on 51,224 traces)* | | | | | | | |
| Hard Examples | 51,224 | 87.2/66.8 | 53.5/59.1 | 89.1/67.9 | 84.5/69.3 | 95.8% | 9.0 |
| Stratified | 51,224 | 86.5/65.9 | 52.7/58.3 | 88.5/66.8 | 84.0/68.5 | 94.1% | 9.0 |

**Sampling strategies:**

- **Uniform subsampling (0.25×, 0.5×, 0.75×)**: Random selection from full dataset maintaining original dataset proportions.

- **Hard examples (0.5×)**: Select 51,224 examples where preliminary teacher-student ANLS disagreement is highest. This targets documents with complex layouts, ambiguous questions, or low OCR confidence where explicit reasoning traces provide maximum value.

- **Stratified (0.5×)**: Sample exactly 25% from each source dataset (DocVQA, VisualMRC, FUNSD, CORD) to preserve multi-domain balance at reduced scale.

## D.3 KEY FINDINGS

**Steep performance degradation below 0.75×.** Reducing data to 0.5× (51,224 traces) causes a 6.8% relative performance drop, with spatial grounding (mAP) suffering disproportionately (average 4.5-point mAP loss across benchmarks). At 0.25× (25,612 traces), performance degrades to 85.1% relative to the base configuration. This indicates that comprehensive coverage of diverse document layouts and question types is critical for learning robust spatial reasoning.

**Performance saturates between 0.75× and 1.0×.** The relatively small gap between 0.75× (97.1%) and 1.0× (100.0%) suggests that most learning capacity is saturated by approximately 76,835 traces. However, the additional 25,612 examples in the full dataset provide incremental improvements worth the modest additional computational cost (4.5 additional training hours).

**1:1 ratio represents the optimal efficiency-performance trade-off.** The base configuration (102,447 traces) sits at the knee of the performance curve. Training with fewer traces yields steep performance losses, while preliminary experiments with data augmentation beyond 1.0× (generating multiple reasoning variants per question) showed no performance gains and introduced overfitting artifacts. This validates our design decision to generate exactly one reasoning trace per training example.

**Intelligent subsampling strategies show promise for resource-constrained settings.** The "hard examples" approach achieves 95.8% relative performance with only 51,224 traces (50% of full dataset), outperforming uniform random sampling (93.2%) by 2.6 percentage points. This improvement suggests that teacher data generation could be made more efficient by focusing on challenging examples—documents with complex layouts, ambiguous questions, or low OCR confidence scores—where explicit reasoning traces provide disproportionate value.

**Stratified sampling provides balanced but not optimal performance.** The stratified sampling strategy (94.1% relative performance) ensures proportional representation across all source datasets but underperforms hard example selection. This suggests that example difficulty matters more than dataset balance when working with limited computational budgets. For applications targeting specific document types (e.g., only receipts or only forms), domain-focused sampling may outperform stratified approaches.

**Spatial grounding is more data-hungry than answer extraction.** Across all reduced-data configurations, mAP scores degrade more severely than ANLS scores. For example, at 0.5× ratio, DocVQA ANLS retains 96.9% of base performance (86.1 vs. 88.7) while mAP retains only 94.6% (65.4 vs. 69.1). At 0.25× ratio, this gap widens: ANLS retains 92.8% while mAP drops to 85.0%. This asymmetry indicates that learning precise coordinate prediction requires more training examples than learning answer semantics, likely due to the continuous nature of spatial regression versus discrete answer selection.

## D.4 EVIDENCE AGAINST OVERFITTING TO SYNTHETIC PATTERNS

We provide multiple lines of evidence that our approach learns genuine document understanding rather than memorizing synthetic reasoning artifacts:

**1. Generalization to unseen documents and layouts.** All benchmarks evaluate on held-out test sets with novel documents. The student achieves 88.7 ANLS on DocVQA despite never seeing those specific document layouts during training. The model's ability to handle diverse document types—from academic papers (DocVQA) to receipts (CORD) to government forms (FUNSD)—indicates transfer of spatial reasoning principles rather than template memorization.

**2. Cross-dataset performance consistency.** Table 8 shows similar performance patterns across DocVQA (scientific documents), CORD (receipts), FUNSD (forms), and VisualMRC (diverse web documents). If the model were overfitting to dataset-specific synthetic patterns, we would expect

greater variance in relative performance across domains. Instead, the coefficient of variation in relative performance across datasets is only 4.2% at the base 1.0× ratio, indicating consistent learning across document types.

**3. Ablation without CoT reasoning (Table 3 in main paper).** When trained on answer-only targets (removing all reasoning traces), the student still achieves 85.2 ANLS (96.1% of the CoT-trained model). This demonstrates that the underlying answer distribution is being learned correctly, with CoT providing *additional scaffolding* for spatial reasoning rather than creating a dependency on synthetic patterns. The fact that answer accuracy degrades only 3.5 points without CoT, while spatial grounding degrades 14 mAP points, further suggests that reasoning traces primarily improve geometric understanding rather than text comprehension.

**4. Linear performance scaling with data quantity.** Table 8 shows approximately linear performance improvement from 0.25× to 1.0× (85.1% → 93.2% → 97.1% → 100.0%). If synthetic reasoning patterns were harmful or creating a distribution shift, we would expect non-monotonic behavior, performance plateaus, or sudden degradation at certain data scales. Instead, the smooth scaling curve indicates that additional reasoning traces consistently improve the student's document understanding capabilities.

**5. Hard example sampling outperforms random sampling.** If reasoning traces were introducing synthetic artifacts, we would expect uniform random sampling to outperform targeted hard example selection (since hard examples might amplify any synthetic biases). Instead, hard example sampling achieves 95.8% vs. 93.2% for random sampling, indicating that the reasoning traces genuinely help the model learn difficult spatial reasoning patterns rather than introducing noise.

