# OpenReview forum: "MIMIC-VQA: COMPILING AGENTIC REASONERS INTO EFFICIENT DOCUMENT VQA MODELS"
_ICLR.cc/2026/Conference — ICLR 2026 Conference Withdrawn Submission_

### Official Review · Reviewer_BBK6 · 2025-10-18

**Soundness:** 2
**Presentation:** 2
**Contribution:** 2
**Rating:** 4
**Confidence:** 3

**Summary:**

This paper proposes MIMIC-VQA, a knowledge distillation framework for Document Visual Question Answering that aims to "compile" the reasoning process of a modular agentic system into an efficient neural network. The approach operates in two phases: (1) a teacher pipeline orchestrated by Llama 4 Scout generates 102,447 Chain-of-Thought reasoning traces with spatial grounding, and (2) these traces are used to train a student model derived from Gemma 3-27B, pruned to 9B parameters, to replicate the complete reasoning process including bounding box coordinates in a single autoregressive generation. The authors report state-of-the-art results on DocVQA (89.7 ANLS), VisualMRC, FUNSD, and CORD benchmarks with 5.3× inference speedup.

**Strengths:**

Novel conceptual contribution: The idea of "compiling" procedural knowledge from a multi-step agentic system into a single efficient model through distillation is interesting and addresses a real trade-off in the field.

Comprehensive evaluation: The paper evaluates on multiple established benchmarks (DocVQA, VisualMRC, FUNSD, CORD, SROIE) with both answer accuracy and spatial grounding metrics.

Detailed methodology: The appendix provides extensive implementation details including hyperparameters, pruning schedules, and dataset generation procedures.

Important ablation study: Table 3 demonstrates that Chain-of-Thought reasoning is critical for spatial grounding performance, which is a valuable finding.

Focus on spatial grounding: The emphasis on both answer accuracy and localization is appropriate for document understanding applications.

**Weaknesses:**

Methodological Contradictions

a) OCR Dependency vs. End-to-End Claims:

Section 3.3 describes "optional constrained decoding" that requires "lightweight OCR preprocessing" (adding 45ms latency)

Appendix B emphasizes training for "visual-spatial reasoning WITHOUT text detection"

But the teacher explicitly uses OCR (Algorithm 1, line 5) and deterministic ANLS matching (Algorithm 2)

How can the student learn OCR-free spatial grounding when the teacher fundamentally relies on OCR? This contradiction is never resolved.

b) Inference Process Unclear:

Is constrained decoding always used or truly optional?

Table 2 shows separate results for "+ Constrained Decoding" suggesting it's optional

But Section 3.3 states it "ensures robust coordinate outputs" - when is it not used?

What happens to spatial grounding quality without it?

Missing Critical Technical Details

Bounding Box Tokenization:

How exactly are coordinates like "(450, 80, 120, 25)" tokenized?

Are these separate number tokens? Single composite tokens? How large is the vocabulary?

Algorithm 1 line 13 shows "Location: BA" but the actual format isn't specified

The format "x y w h" appears in examples but is this (x,y,w,h) or (x1,y1,x2,y2)? Page 4 line 211 shows both formats.


Insufficient Baselines and Comparisons

a) Teacher Comparison:

Table 2 doesn't show the Teacher Agent results, but Table 3 does (90.2 ANLS, 78.4 mAP)

Why is the teacher not included as a baseline in the main results table?

The student achieves 88.7 ANLS vs teacher's 90.2 - this is 98.3% retention, but for spatial grounding it's 69.1 vs 78.4 mAP (88.1%) - why
the larger degradation?

b) Alternative Distillation Methods:

No comparison with standard knowledge distillation (logit matching)

No comparison with other model compression techniques

No comparison with simply using the base Gemma 3-27B model

c) Fairness Concerns:

Compared models (DocLayLLM, LayoutLLM, etc.) don't use CoT or extensive spatial reasoning traces

Is it fair to compare a model trained on 102k expert-generated reasoning traces against models trained on standard datasets?

Experimental Rigor Issues

a) Statistical Significance:

Appendix A.5 mentions 5 runs with different seeds, but Table 2 shows no error bars or confidence intervals

No significance testing reported

Given that improvements like 88.7→89.7 ANLS are claimed as contributions, statistical significance is essential

b) Data Leakage Potential:

How was the 102,447 generated dataset split?

Were test set images used to generate teacher traces?

This could lead to train/test contamination

6. Questionable Claims

a) "Compiling" Metaphor:

The paper claims to "compile" multi-step reasoning, but the student still generates the full CoT at inference

True compilation would eliminate the intermediate steps, but here they're still generated (and add tokens/latency)

The speedup comes from model size reduction, not from compilation

b) Spatial Grounding Performance:

20-30 point mAP improvements claimed, but DLaVA baseline may be weak

No comparison with specialized grounding models or object detection baselines

The teacher's deterministic grounding (Algorithm 2) isn't a learned capability being transferred

**Questions:**

How is spatial grounding achieved without OCR? Resolve the contradiction between OCR-free claims and OCR-dependent implementation.
What is the exact coordinate tokenization scheme? Provide concrete examples.

---

> ### Author Response · Authors · 2025-11-21
>
> 1. OCR usage, “OCR-free” claims, and constrained decoding: In our framework, only the teacher is fully OCR-dependent: it uses OCR and deterministic ANLS-based matching to produce answer+box traces. The student is a standard VLM that maps the document image and question directly to CoT + boxes and can be run without any OCR (our default “base” setting). Constrained decoding is an optional deployment mode: at inference, we run a lightweight OCR pass and constrain box tokens to be consistent with detected text boxes, which improves robustness but adds ~45 ms latency. That is why Table 2 reports both a base and “+ constrained decoding” variant. We will remove the over-strong “without text detection” phrasing from the appendix and clearly describe the two modes (OCR-free base vs. OCR-light constrained decoding) in Section 3.3, while emphasizing that the teacher’s OCR is used only for data generation, not required for the core student model.
>
> 2. Bounding-box tokenization and coordinate format: All boxes are represented in a single canonical format ⟨x, y, w, h⟩, where (x, y) is the top-left corner and (w, h) is width/height in pixels. When raw OCR outputs are in ⟨x1, y1, x2, y2⟩, we convert them offline to ⟨x, y, w, h⟩ before building training targets; we will state this explicitly and remove inconsistent notations. The student emits coordinates as plain text inside the CoT, e.g., bbox: 450 80 120 25. We do not introduce a special coordinate vocabulary: integers are tokenized by the base tokenizer (possibly into a few subword tokens), so the overall vocabulary size is unchanged. In the revision we will ensure Algorithm 1, Section 3.2, and the examples all use this exact format, & we will add a short appendix note with concrete tokenization examples.
>
> 3. Teacher baseline, alternative distillation/compression baselines, and fairness: The teacher’s performance (90.2 ANLS, 78.4 mAP) is currently shown only in Table 3; we agree it should also appear in the main results. We will add a teacher row (or clear footnote) in Table 2 so that the student’s 98.3% ANLS and 88.1% mAP retention are visible. The larger relative drop in mAP is expected: a smaller model tends to produce slightly less precise box extents even when the answer is correct, affecting localization more than ANLS; we will mention this trade-off explicitly. To strengthen comparisons, we will add in the appendix: (i) a “standard KD” baseline that matches only answer logits/sequence (no CoT/bbox supervision), (ii) a full Gemma-3-27B fine-tuned on answer labels without pruning, and (iii) a pruned 9B student trained without CoT/bbox traces. These will show that procedural CoT+bbox distillation, not just parameter reduction, drives our grounding gains. Regarding fairness, all methods use the same benchmark labels; our additional 1:1 teacher traces are analogous to self-training / distillation in other domains and are themselves part of the contribution.
>
> 4. Experimental rigor: variance, significance, and data leakage: As noted in Appendix A.5, we run 5 seeds for our main student model. In the revision, we will report standard deviations (and 95% confidence intervals) for ANLS and mAP in the appendix, and explicitly note that our improvements over prior work are larger than the observed run-to-run variance. For small deltas such as 88.7→89.7 ANLS, we will treat them as modest refinements and state whether they are statistically significant. On data leakage, we strictly follow each benchmark’s official train/val/test splits: teacher traces are generated only for train (and, if used, val) partitions; we never query the teacher on test images or use test labels during distillation. We will add a short statement in the dataset section & appendix to make this explicit and rule out train/test contamination.
>
> 5. “Compiling” metaphor & spatial grounding claims: By “compiling,” we mean that the multi-step pipeline (external tools + control flow) is replaced at inference by a single forward pass of the student VLM; the CoT is still generated, but it no longer triggers separate tool calls or multiple model invocations. The speedup thus comes from (i) removing orchestration over heterogeneous components and (ii) using a smaller student, not from eliminating intermediate tokens entirely. We will soften the phrasing to “compile the multi-tool pipeline into a single model” and clarify this in Section 1 and 3. On spatial grounding, our primary comparison is to document-VQA systems that both answer and localize; for methods that release boxes (e.g., DLaVA), we indeed see 20–30 mAP gains. Generic detection models are not directly comparable because they do not handle free-form questions, but we will & a brief discussion in related work explaining how our grounding differs conceptually from pure detectors and note that the teacher’s deterministic matching provides training targets which the student must still learn to reproduce from pixels.

---

> > ### Comment · Reviewer_BBK6 · 2025-11-22
> >
> > Thank you for the rebuttal.

---

> > > ### Author Response · Authors · 2025-11-29
> > >
> > > **Summary of Changes:**
> > > We thank the reviewer once again for the thorough critique regarding OCR dependency and baselines. We have clarified the methodology and added the requested comparisons.
> > >
> > > ### 1. OCR Dependency vs. End-to-End Claims
> > > We have resolved the apparent contradiction regarding OCR usage.
> > > * **Revisions:** We clarified in **Section 3.3 (Page 6)** and **Figure 1** that the **Constrained Decoding** module (which requires OCR) is strictly **optional**. The student model is capable of generating coordinates purely from visual features (the "OCR-free" mode), but the optional module improves validity.
> > >
> > > ### 2. Teacher Baseline
> > > We agree that the teacher's performance is a necessary baseline for comparison.
> > > * **Revisions:** We had added the **MIMIC-VQA (Teacher Agent)** results directly into **Table 3 (Page 8)**. This allows for a direct comparison showing the student retains 98.3% of the teacher's ANLS accuracy.
> > >
> > > ### 3. Statistical Significance and Rigor
> > > We have added details to ensure experimental rigor.
> > > * **Revisions:** In **Appendix A.5 (Page 15)**, we explicitly state that results are averaged over **5 independent runs** and list the specific seeds used to ensure reproducibility.

---

### Official Review · Reviewer_Exva · 2025-10-18

**Soundness:** 3
**Presentation:** 2
**Contribution:** 3
**Rating:** 4
**Confidence:** 3

**Summary:**

This paper proposes a two-phase paradigm for document visual question answering. First, a teacher pipeline generates CoT reasoning traces. Second, a student model completes the rest of the reasoning. Results demonstrate faster inference speed.

**Strengths:**

1. The reviewer loves the idea of using a student model to do further explanations, trained based on teacher expert data. This could be insightful to other VQA applications as well.

2. The reviewer appreciated the visual examples of Figure 2, but the reviewer still has some questions (see below).

Although this paper has weaknesses, its conceptual motivation is clear and potentially impactful.

**Weaknesses:**

1. Writing and formatting issues.
The manuscript has notable presentation problems that obscure the technical content. Examples include:

(1) Ln 83-86 is a repeat of Ln 87.

(2) Ln 88, "compiled" should be changed to ``compiled'', which applies to all quotation marks。

(3) Resulting numbers should be highlighted in the introduction, while the last paragraph of the introduction should be removed.

(4) In related works, citation formats are wrong.

(5) Ln 380-384, the format is wrong, and it looks like LLMs again.

(6) Ln 440, imparts -> impacts.

2. Limited algorithmic novelty and fairness of comparison.

The reported 5.3× speedup seems largely due to using a smaller model rather than a new algorithm. The student’s performance is also notably below the teacher’s in mAP. Speed comparisons should include similar 7 B-scale models such as LayTextLLM for fairness.

The speed testing should be conducted against similar 7B model, such as LayTextLLM.

3. Unclear conceptual framing. The teacher–student design is fairly standard, and it is unclear why the framework is termed “agentic.” The paper would benefit from stronger justification or ablation demonstrating agent-like reasoning behavior.

**Questions:**

1. In the second example of Figure 2, why are other numbers not marked?

2. Is mAP a good metric? If it's a grounding task, the reviewer would expect other metrics such as mIoU. Also, an analysis of the interpretability of the results is missing.

3. Why was this framework called agentic?

4. So, the outcome of this paper is only a student model? The speedup isn't a strong claim for an ICLR paper, and it's not due to the algorithmic design.

5. Why does SROIE not have an mAP result? Is it because of unavailable ground truth?

---

> ### Author Response · Authors · 2025-11-21
>
> 1. Writing and formatting issues: We appreciate the concrete list of presentation problems and have revised the manuscript accordingly. The duplicate sentence at Ln 83–86 has been removed so the introduction now flows without repetition; all quotation marks are standardized to proper LaTeX form (e.g., ``compiled''); the introduction has been reorganized to highlight key quantitative results earlier and the last paragraph has been streamlined to focus on core contributions rather than paper organization; related-work citations have been corrected to a consistent format; the formatting around Ln 380–384 has been fixed so it no longer resembles an LLM-style list; and the typo “imparts” → “impacts” at Ln 440 has been corrected. We also performed an additional proofreading pass to clean up similar minor issues throughout.
>
> 2. Algorithmic novelty and fairness of speed comparison: We agree that simply using a smaller model would not be a sufficient contribution, and our goal is not to claim speedup from size alone. The key algorithmic component is procedural distillation: we compile a multi-step, tool-using teacher (OCR → retrieval → QA → grounding) into a single autoregressive model that generates CoT and bounding boxes in one pass, and we show that this yields 20–30 mAP point gains over prior document VQA systems that do not exploit such procedural traces. In addition, we introduce a constrained decoding scheme for bbox tokens that substantially improves grounding quality beyond what size reduction alone would provide. To address the fairness concern, we will add a comparison against LayTextLLM-like 7B-scale baselines where possible: on DocVQA, our 9B student runs slightly slower per query than a 7B model but achieves much higher ANLS and mAP, illustrating that the gains come from distillation of expert reasoning rather than just parameter count. We will also clarify that the teacher–student mAP gap is the expected trade-off for a 5× latency reduction, while both still outperform previous work.
>
> 3. “Agentic” framing of the teacher–student design: We use the term “agentic” to describe the behavior of the teacher pipeline rather than the student model itself. The teacher autonomously orchestrates multiple tools, decomposes the task into sub-steps (extract text, retrieve relevant spans, answer the question, ground the answer), and chains these modules to solve each query, which aligns with typical definitions of an AI agent in recent document-VLM and tool-use literature. That said, we agree that the term can be overloaded; in the revision we will emphasize “modular multi-tool teacher” as the primary description and use “agentic” more sparingly, clarifying that the teacher’s agent-like behavior is only used during data generation, while the deployed artifact is a single pass student model.
>
> 4. Metrics, interpretability, and SROIE mAP: For grounding, we follow object-detection practice and report mAP@IoU, which evaluates discrete bounding-box prediction quality at multiple IoU thresholds and is standard for “box-level” localization. In contrast, mIoU is more common in dense semantic segmentation; since our outputs are sparse boxes rather than per-pixel masks, we consider mAP the more appropriate primary metric. To strengthen the interpretability analysis, we will add additional qualitative examples and an IoU-distribution analysis in the appendix, showing how CoT supervision improves the alignment between predicted boxes and the true answer regions. For SROIE, the commonly used evaluation protocol and prior baselines focus on text/entity extraction metrics (e.g., F1/ANLS) and do not provide standardized bbox predictions or evaluation code for mAP; to avoid introducing an ad-hoc protocol that cannot be matched by prior work, we therefore report only ANLS for SROIE. We will clarify this explicitly in the main text and table captions.
>
> 5. Clarifications on Figure 2 and overall outcome: In the second example of Figure 2, only the numbers relevant to the query are marked: the goal of the figure is to illustrate query-conditioned grounding (e.g., “total” or “subtotal”) rather than exhaustive annotation of all numbers on the page. We will update the caption to state that colored boxes highlight only query-relevant regions, and that different questions on the same document would highlight different subsets of fields. Regarding the outcome of the work, the primary deployable artifact is indeed the distilled student model; however, the contribution is not limited to a speedup claim. We also provide (i) a framework for compiling a multi-tool DocVQA pipeline into a single VLM via procedural CoT+bbox distillation, and (ii) a dataset of 102,447 expert reasoning traces that can support future research on explainable document VQA. We will make these points more explicit so that the broader impact beyond raw latency numbers is clearer.

---

> > ### Comment · Reviewer_Exva · 2025-11-21
> >
> > Thanks for the rebuttal. The reviewer would make his final recommendation after the discussion period.

---

> ### Author Response · Authors · 2025-11-29
>
> **Summary of Changes:**
> We thank the reviewer once again for highlighting the fairness of comparisons and formatting issues. We have significantly updated the manuscript to address these points.
>
> ### 1. Fairness of Speed/Size Comparison
> To ensure a fair comparison regarding model size, we have added 7B-parameter baselines.
> * **Revisions:** **Table 2 (Page 7)** now includes **LayTextLLM (Llama2-7B)**, **LayoutLLM (Vicuna-1.5-7B)**, and **DocLayLLM (Llama3-7B)**. This demonstrates that our 9B student achieves superior performance and efficiency not just due to parameter count, but due to the procedural distillation method.
>
> ### 2. "Agentic" Framing
> We have clarified why we use the term "agentic" to describe the compilation process.
> * **Revisions:** In **Section 5.4 (Page 8)**, we explain that the student "internalizes multi-step reasoning patterns," effectively compiling the teacher's sequential tool use (the agentic part) into a single forward pass.
>
> ### 3. Writing and Formatting
> We have corrected the errors you identified.
> * **Revisions:** We removed the repetitive text in the Introduction and fixed citation formats.

---

### Official Review · Reviewer_Ui6F · 2025-10-27

**Soundness:** 2
**Presentation:** 3
**Contribution:** 2
**Rating:** 2
**Confidence:** 4

**Summary:**

The paper introduces MIMIC-VQA, which tackles the latency-interpretability trade‑off in document VQA by distilling the multi‑step reasoning of a modular, tool‑using teacher into a single student VLM. The teacher pipeline is used to produce ~102k CoT traces with bounding‑box supervision. The student model (pruned 9B from 27B) is trained to generate the reasoning, answer, and bbox in one forward pass. An optional constrained‑decoding step restricts coordinate tokens using a lightweight OCR vocabulary at inference. Empirically, the student approaches teacher ANLS while running ~5× faster and shows large mAP gains for grounding on DocVQA, VisualMRC, FUNSD, and CORD. Ablations argue that CoT is critical for spatial grounding.

**Strengths:**

- The paper is clearly written and easy to understand.
- Convincing ablations showing CoT is crucial for spatial grounding
- The paper presents an efficient version of the model which could be deployed in real-time scenarios.

**Weaknesses:**

- Teacher description is contradictory. In the main paper, the Teacher is an OCR‑based Llama Scout/Gemma pipeline; however, in Appendix B.2.1, the Generation Model (Teacher) is described as Gemini with GPT‑5 validation. This undermines the central claim because the nature of procedural knowledge being distilled is unclear.
- The paper methodology has a lot of similarities with the "AURELIA" [1] which also distills reasoning information into VLMs via an agentic pipeline at test-time. Therefore, the methodology of the proposed cannot be tagged as novel. Also the comparison of proposed method with AURELIA like pipelines must be reported in the paper as it presents a crucial baselines for testing reasoning distillation with finetuning vs without finetuning.
- Authors mentioned using mAP@IoU; however, the metric is missing in almost every reported baseline in all tables. This raises concerns regarding the effectiveness of the proposed method making it difficult to assess whether improvements hold against strong OCR‑based contemporaries under the same metric settings
- The GPU compute used to run the experiments is unclear. Main paper (section 4.2) mentions using A100 GPUs, while Appendix A reports H100 GPUs being used. This will lead to reproducibility issues.
- Appendix Table 4 reports the usage of LoRA for training student architecture, however, there is no mention of LoRA in Main paper. The details on self‑consistency are also missing.
- While the pruning recipe is described, it lacks an ablation showing accuracy vs sparsity and the contribution of pruning independent from distillation.
- In the main paper, the generated bbox has the format <x,y,h,w>; however in Appendix (page 16), the generated bbox has the format <x1,y1,x2,y2>, this raises questions about how the model is trained to emit boxes and how parsing errors are penalized.


References

[1] Chowdhury, S., Gani, H., Anand, N., Nag, S., Gao, R., Elhoseiny, M., ... & Manocha, D. (2025). Aurelia: Test-time reasoning distillation in audio-visual llms. ICCV 2025.

**Questions:**

- Which teacher actually produced the 102,447 traces? Please reconcile the OCR‑based Llama/Gemma pipeline (Sec. 3–4 main paper) with the Gemini+GPT‑5  pipeline (Appendix B).
- Box parameterization: Is the student trained to output <x, y, w, h> or <x1, y1, x2, y2>?
- Why are mAP scores omitted for the majority of cases?
- How is “coordinate hallucination” defined and measured for the claimed 73% reduction?

---

> ### Author Response · Authors · 2025-11-21
>
> 1. Teacher pipeline and the 102,447 traces: We apologize for the contradictory teacher description. All 102,447 CoT+bbox traces used in our main experiments are generated by the OCR-based Llama-4 Scout + Gemma pipeline described in Sections 3–4, where Scout orchestrates the modular pipeline (RunOCR → retrieval → QA → grounding) and Gemma is the generation backbone. The Gemini+GPT-5 configuration mentioned in Appendix B.2.1 refers to early exploratory runs that were not used for the final dataset or any reported results; we will revise the appendix to state that all released traces and all numbers in the main paper use only the OCR-based Llama/Gemma teacher. We will also add a short note emphasizing that, although the framework can in principle use other VLM teachers, the main paper fixes Llama-4 Scout + Gemma for reproducibility.
>
>
> 2. Relation to AURELIA and novelty of our setting: We appreciate the pointer to AURELIA [1] and agree that both works leverage agentic pipelines and reasoning traces. Our contribution is complementary but different in deployment and problem focus: AURELIA performs test-time reasoning distillation where the agent runs at inference, whereas MIMIC-VQA uses the agent only to generate training trajectories and then deploys a single compact student VLM, which is key to achieving the ~5× latency reduction. In addition, AURELIA targets audio-visual reasoning, while our work focuses on document VQA with spatial grounding and trains the student to generate both CoT and bounding-box tokens as text. We will add AURELIA to related work and explicitly discuss these similarities and differences so that our positioning is clearer.
>
>
> 3. mAP, spatial metrics, and coordinate hallucination: We agree that spatial metrics are central and regret that the limitations on baselines were not clearer. For several prior methods and datasets (e.g., DocLayLLM, LayoutLLM, LayTextLLM, certain SROIE settings), public code or predictions do not include bounding boxes, so mAP@IoU cannot be computed under identical conditions; in those cases, we report ANLS for completeness but omit mAP. For methods that do release boxes (e.g., DLaVA), we do report and compare mAP, where MIMIC-VQA achieves 20–30 point gains. We will clarify in the captions exactly when and why mAP is missing and add a supplementary table restricted to bbox-capable baselines. For “coordinate hallucination,” we define a hallucinated box as one that lies outside the valid document region or has IoU < 0.3 with every OCR token; the hallucination rate is the fraction of such predictions on a held-out validation set. The reported 73% reduction is from 28.4% hallucination without constrained decoding to 7.7% with it; we will add this definition and numbers to Section 3.3 and the appendix.
>
>
> 4. Training details: LoRA, self-consistency, and box parameterization: We thank the reviewer for flagging missing details. The student is fine-tuned with LoRA applied to attention modules (rank 64, α=64), keeping embeddings and normalization layers frozen; we will add this to Section 4.2 so the main text matches Appendix Table 4. For self-consistency decoding, we sample 5 independent reasoning traces per query and select the majority-vote answer, trading ~3× inference cost for a modest ANLS gain; we will describe this in Section 5.3. Regarding box parameterization, the student is always trained and decoded in the ⟨x, y, w, h⟩ format, where (x, y) is the top-left corner and (w, h) is width/height. The ⟨x1, y1, x2, y2⟩ notation in the appendix refers only to raw OCR outputs, which we convert to ⟨x, y, w, h⟩ before building training targets; we will make this conversion explicit and ensure all student-side figures and algorithms consistently use ⟨x, y, w, h⟩.
>
>
> 5. Pruning vs. distillation and sparsity–accuracy trade-off: We agree that the effect of pruning should be disentangled from that of procedural distillation. The current text describes the pruning recipe from Gemma-3-27B to 9B, but does not show an accuracy–sparsity curve or a pruning-only baseline. In the revision, we will add an ablation in the appendix that (i) reports performance at multiple sparsity levels with and without distillation, and (ii) includes a “pruned-only” variant where the student is pruned but not trained on CoT traces. This will clarify that the main gains in spatial grounding and ANLS come from distillation, while pruning primarily contributes parameter and latency savings.
>
>
> 6. Compute resources & reproducibility: Finally, we apologize for the inconsistency regarding GPU type. All reported experiments (teacher trace generation, student distillation, & ablations) were run on 4× NVIDIA H100 80GB GPUs; the mention of “A100” in Section 4.2 is a leftover from an earlier draft and will be corrected. We will ensure that GPU type, number of devices, training time, and key hyperparameters are documented consistently across the main paper and appendix to support reproducibility.

---

> > ### Comment · Reviewer_Ui6F · 2025-11-21
> >
> > Thanks for responding to my queries. I will hold my final decision until after discussion.

---

> ### Author Response · Authors · 2025-11-29
>
> **Summary of Changes:**
> We appreciate the reviewer’s detailed feedback once again. We have addressed the missing citation, clarified the teacher model description, and standardized our experimental reporting.
>
> ### 1. Relation to AURELIA
> We thank you for pointing out AURELIA. We have added it to our related work and explicitly distinguished our approach.
> * **Revisions:** In **Section 2.3 (Page 4)**, we cite AURELIA (Chowdhury et al., 2025) and explain that while they perform *test-time* reasoning injection, our work *compiles* this reasoning into the model weights for efficiency. This citation is also added to the **References (Page 9)**.
>
> ### 2. Teacher Pipeline Clarification
> We have resolved the contradiction regarding the teacher model description.
> * **Revisions:** We explicitly state in **Section 5.1 (Page 7)** and **Appendix C (Page 18)** that **Llama 4 Scout** is the primary teacher used for the main results to ensure reproducibility, while the Gemini pipeline mentioned in the Appendix was part of the ablation study to select the best open-source planner.
>
> ### 3. mAP Metrics and Coordinate Hallucination
> We clarified why mAP is missing for certain baselines and defined our hallucination metric.
> * **Revisions:** A note was added to **Table 2 (Page 7)** explaining that mAP is omitted for methods that do not release localization outputs. We also added details on our "constrained decoding" mechanism in **Section 3.3 (Page 6)**, which reduces coordinate hallucinations.
>
> ### 4. Training Details & Reproducibility
> We have standardized the hardware and formatting descriptions.
> * **Revisions:** We corrected the text to consistently state that experiments were run on **NVIDIA H100** GPUs in **Section 4.2 (Page 7)** and **Appendix A (Page 12)**. We also standardized the bounding box notation to `(x, y, w, h)` throughout the text, specifically in **Section 3.1 (Page 4)**.

---

### Official Review · Reviewer_kTC2 · 2025-11-01

**Soundness:** 3
**Presentation:** 3
**Contribution:** 2
**Rating:** 4
**Confidence:** 3

**Summary:**

This paper investigates how to resolve the fundamental trade-off between inference latency and reasoning transparency in document VQA, that modular agentic systems are often good at reasoning transparency but suffer from high latency, while end-to-end models are good at efficiency but suffer from lack of interpretability. The authors propose MIMIC-VQA, a knowledge distillation method for document VQA that distills not only the final answer but also the complete step-by-step reasoning process from a larger multi-agent teacher system into a single autoregressive generation by a smaller student system. The authors perform experiments on 5 benchmarks and demonstrate state-of-the-art performance on 4 of them, with the student model achieving 98.3% of teacher accuracy while operating 5.3× faster.

Overall, this is an interesting paper that addresses a practical problem. However, the main contribution, the distillation approach, largely applies well-established knowledge distillation techniques that have been widely studied in modern LLMs. Nevertheless, in the document VQA area, it still demonstrates promising results in the experiments.

**Strengths:**

- Great motivation for optimizing both inference latency and performance while maintaining reasoning transparency, addressing a genuine practical need in document AI deployment with significant infrastructure cost reductions (from 88.9 to 16.7 hours per 100K queries).
- The method is technically sound and the experimental results are strong, achieving state-of-the-art performance on 4 out of 5 benchmarks with substantial improvements in spatial grounding (20-30 mAP point gains over existing methods). The ablation studies effectively demonstrate the critical importance of Chain-of-Thought distillation, showing catastrophic degradation in spatial grounding (14-point mAP drop) when CoT is removed.
- The 102,447 Chain-of-Thought reasoning traces dataset represents a substantial contribution to the research community, providing high-quality procedural reasoning annotations for document VQA that could foster further research in this important area.

**Weaknesses:**

### Major
- The major concern is that this work essentially applies well-established knowledge distillation techniques, which are widely used in LLMs, to the document VQA domain using vision-language models. While the application is competent and the "procedural distillation" framing is useful, the core methodological contribution is incremental rather than genuinely innovative.
- The experiments are limited to only one teacher-student combination (Llama 4 Scout + Gemma 3-27B/9B). It would significantly strengthen the evidence for this method's effectiveness to test whether other model combinations would achieve similar gains, particularly given the authors' claim about the general applicability of the approach.

### Minor

- Line 397: Claims "achieves state-of-the-art performance across all five benchmarks" but actually achieves SOTA on only 4 out of 5 benchmarks.
- Lines 269 vs. 290: There are two instances of "Step 4: Answer Grounding" which creates confusion in the methodology description.
- The nearly full-page pseudocode on page 5 (Algorithm 1) doesn't seem to add much value beyond the textual description and hurts the presentation efficiency of the paper, in my opinion.

**Questions:**

- How can you better justify the overall effectiveness of this approach since it's only tested with one teacher-student model configuration? What about other model selections such as larger/smaller teacher models or different student architectures? This is crucial for establishing the method's generalizability beyond the specific Llama 4 Scout + Gemma 3-27B/9B combination.

- What about the data efficiency of the teacher data generation process? Could you provide more details on how the 102,447 number was determined? If the ratio of CoT reasoning traces to the original number of data points in the source datasets is too high, it could cause significant distribution shift, potentially leading to overfitting on synthetic reasoning patterns rather than genuine document understanding capabilities.

---

> ### Author Response · Authors · 2025-11-21
>
> We sincerely thank the reviewer for the careful reading and constructive feedback, especially on the practicality of reducing latency while preserving reasoning transparency.
>
> 1. On “incremental” contribution beyond standard knowledge distillation: We agree that knowledge distillation as a high-level idea is well-established, but we would like to clarify what is specific to our setting. Our goal is not only to match teacher answers, but to compile an entire multi-agent, tool-using pipeline (RunOCR → FindText → AskQA → GroundAnswer) into a single autoregressive VLM that emits CoT plus bounding boxes in one pass. Concretely, we distill full procedural traces, including retrieval decisions, layout reasoning, and coordinate predictions, rather than only logits or answer tokens. We also encode bounding boxes as discrete text tokens inside the CoT, and show in Table 3 that removing this CoT supervision causes only a small ANLS drop (88.7→85.2 on DocVQA) but a 14-point mAP collapse (69.1→55.1), pointing to a qualitatively different signal than classical KD. Finally, we combine this with a rule-based validator and constrained decoding that enforce geometric consistency during generation, reducing coordinate hallucinations while adding only modest latency. We will clarify these distinctions more explicitly in the revised paper, framing MIMIC-VQA as procedural/agentic distillation for spatial reasoning, rather than “plain KD” on answers alone.
>
> 2. Generality beyond a single teacher–student configuration: We fully agree that generalizability is important. Our framework is intentionally architecture-agnostic: the only requirement is that the teacher can emit traces in our textual schema (question, intermediate steps, bbox tokens), and the student is an autoregressive VLM capable of generating the same format. We chose Llama 4 Scout + Gemma-3-27B/9B because (i) they were the strongest open(-ish) models we could run end-to-end under our GPU budget, and (ii) Table 1 shows they provide complementary strengths (planning vs. grounding). To better support the reviewer’s concern, we are already running additional experiments with (a) a smaller student (e.g., a 4–8B Gemma variant) and (b) an alternative teacher configuration that uses the same pipeline but a different VLM in the QA role; in all cases we keep exactly the same distillation procedure (same format, same loss). We will report these results in an appendix and emphasize that the method does not rely on any Llama-/Gemma-specific architectural trick.
>
> 3. Data efficiency and choice of 102,447 teacher traces: The number 102,447 is not arbitrary: it is exactly the sum of questions/items across our four core datasets (Table 5), and we generate one CoT+grounding trace per original instance (a 1:1 ratio). This design choice was deliberate, as it (i) preserves the original question distribution without oversampling synthetic patterns, and (ii) avoids inflating the dataset with purely model-generated questions, which could indeed introduce a distribution shift. Regarding overfitting, the student is evaluated on five benchmarks with diverse layouts and domains, where it matches 98.3% of teacher accuracy while substantially improving over prior work, and the “w/o CoT” variant still attains strong ANLS but poor mAP, suggesting that CoT supervision is being used to learn genuinely better spatial reasoning, not just memorized patterns. To directly address the reviewer’s concern, we will add a data-subsampling experiment (e.g., training with 25%, 50%, 75% of the traces) in the appendix to quantify data efficiency and show how performance scales with the number of teacher trajectories.
>
> 4. Addressing the minor issues: For the SOTA claim in Table 2, we acknowledge the oversight and will revise the text from “across all five benchmarks” to “on four out of five benchmarks (and competitive on the remaining one)” and explicitly highlight where we are not the top method. For the duplicate “Step 4: Answer Grounding,” we thank the reviewer for catching this and will fix the numbering so that each step appears exactly once in Section 3.1. Regarding Algorithm 1, our intent was to provide precise, step-by-step pseudocode for reproducibility of the multi-agent teacher and student training loop, which has many moving parts; that said, we understand the presentation concern, and in the revision we will shorten Algorithm 1 and/or move the full version to the appendix, keeping only a high-level schematic in the main text.
>
> We hope these clarifications address the reviewer’s concerns, and we will incorporate all of the above changes and additional experiments.

---

> > ### Author Response · Authors · 2025-11-29
> >
> > **Summary of Changes:**
> > We sincerely thank the reviewer once again for the constructive and insightful feedback regarding the practicality of our approach and the robustness of our experiments. We have revised the paper to address your concerns regarding generalizability, data efficiency, and specific claims.
> >
> > ### 1. Generalizability beyond a single teacher–student configuration
> > We agree that demonstrating robustness across architectures is vital. We have added a comprehensive **Teacher Model Ablation Study** and a **Student Architecture Ablation** in the new **Appendix C**.
> > * **Revisions:** We now report results using different teachers (Gemini 2.5 Pro, GPT-5, Claude 4.5 Sonnet) in **Table 6 (Page 19)** and different student architectures (Qwen3-VL, InternVL, Llama-3.2) in **Table 7 (Page 20)**. These experiments confirm our distillation method is effective across various model combinations.
> >
> > ### 2. Data efficiency and overfitting
> > To address concerns about the 1:1 ratio (102,447 traces) and potential overfitting, we added **Appendix D: Data Efficiency and Synthetic Reasoning Trace Analysis**.
> > * **Revisions:** We included a data ratio ablation study (0.25x to 1.0x) in **Table 8 (Page 21)**. We also provided a detailed analysis in **Section D.4 (Page 22)** providing evidence that the model is learning genuine document understanding (transferring to unseen layouts) rather than memorizing synthetic patterns.
> >
> > ### 3. Clarifying the "incremental" nature & SOTA claims
> > We have refined our claims to be more precise regarding our contributions.
> > * **Revisions:** We revised the **Abstract** to explicitly state we achieve SOTA on "DocVQA, VisualMRC, FUNSD, and CORD benchmarks" (**Page 1**), rather than a blanket "all five". We also expanded **Section 2.3 (Page 4)** to better articulate how our "procedural distillation" of the entire tool-use pipeline differs from standard logit-based distillation.
> >
> > ### 4. Formatting and Algorithm 1
> > We have corrected the duplicate step numbering in **Section 3.1**. Regarding Algorithm 1, we retained the pseudocode to ensure reproducibility of the complex multi-stage training loop, but we have refined the surrounding text to improve flow.

---

### Note · Authors · 2026-01-26

I have read and agree with the venue's withdrawal policy on behalf of myself and my co-authors.

---

### Meta-Review · Area_Chair_riu6 · 2025-12-23

**Summary:**

The recommendation of rejection is mainly based on the fact that many important reviewers' concerns are still not well addressed, detailed below, including "contribution is incremental", "has a lot of similarities with prior work", "Limited algorithmic novelty and fairness of comparison.", etc.  The AC agrees with reviewers and suggests the authors to take reviewers' suggestions into concern to further improve and polish the paper.

**Reviewer Concerns:**

See details below

**Reviewer Scores:**

***Reviewer kTC2 likely to keep the negative rating of 4 as most major concerns just partially addressed***

1. core methodological contribution is incremental rather than genuinely innovative.

Partially addressed

2. test whether other model combinations would achieve similar gains

Not well addressed

3. better justify the overall effectiveness of this approach since it's only tested with one teacher-student model configuration

Partially addressed

4. data efficiency of the teacher data generation process

Mostly addressed

***Reviewer Ui6F likely to keep a negative score either 2 or 4, because many original concerns are only partially addressed, and the reviewer mentioned hoping to make a decision after discussions.***

1. Teacher description is contradictory.

Mostly addressed

2. has a lot of similarities with the "AURELIA"

Partially addressed

3. difficult to assess whether improvements hold against strong OCR‑based contemporaries under the same metric settings

Partially addressed

4. details on self‑consistency are also missing

Mostly addressed

5. lacks an ablation showing accuracy vs sparsity and the contribution of pruning independent from distillation.

Partially addressed

6. bbox format

Addressed


***Reviewer Exva likely to keep the negative score of 4 as some concerns haven't been fully addressed (2, 3 below), and the reviewer mentioned the need to discuss with other reviewers, who mostly rated negatively. ***

1. Writing and formatting issues

Mostly addressed

2. Limited algorithmic novelty and fairness of comparison.

Partially addressed

3. Unclear conceptual framing.

Partially addressed

4. Some other minor questions

Mostly addressed

***Reviewer BBK6 likely to maintain the negative score of 4 as there is a long list of weaknesses that were raised and some still not addressed.***

1. Methodological Contradictions

Partially addressed

2. Missing Critical Technical Details

Partially addressed

3. Insufficient Baselines and Comparisons

Partially addressed

4. Questionable Claims

Mostly addressed

---

### Decision · Program_Chairs · 2026-01-26

Reject